# MeCP2 binds to methylated DNA independently of phase separation and heterochromatin organisation

Raphaël Pantier [1], Megan Brown[1], Sicheng Han[1], Katie Paton[1], Stephen Meek[2], Thomas Montavon[3], Nicholas Shukeir[3], Toni McHugh[1], David A. Kelly[1], Tino Hochepied[4,5], Claude Libert [4,5], Thomas Jenuwein [3], Tom Burdon[2] & Adrian Bird [1] ✉

Correlative evidence has suggested that the methyl-CpG-binding protein MeCP2 contributes to the formation of heterochromatin condensates via liquid-liquid phase separation. This interpretation has been reinforced by the observation that heterochromatin, DNA methylation and MeCP2 co-localise within prominent foci in mouse cells. The findings presented here revise this view. MeCP2 localisation is independent of heterochromatin as MeCP2 foci persist even when heterochromatin organisation is disrupted. Additionally, MeCP2 foci fail to show hallmarks of phase separation in live cells. Importantly, we find that mouse cellular models are highly atypical as MeCP2 distribution is diffuse in most mammalian species, including humans. Notably, MeCP2 foci are absent in *Mus spretus* which is a mouse subspecies lacking methylated satellite DNA repeats. We conclude that MeCP2 has no intrinsic tendency to form condensates and its localisation is independent of heterochromatin. Instead, the distribution of MeCP2 in the nucleus is primarily determined by global DNA methylation patterns.

MeCP2 is an epigenetic regulator which controls gene expression by recognising both methylated 'CG' and 'CA' sequences[1–9]. MeCP2 function is particularly important in neurons[10–14] and its mutation in humans is responsible for Rett syndrome, a severe neurological disorder primarily affecting females[15]. Experiments in cell lines and transgenic mice indicate that MeCP2 acts as a molecular 'bridge' between methylated DNA and the co-repressor complex NCoR to restrain expression of large numbers of neuronal genes[16–18]. This model is supported by the observation that missense mutations in the methyl-DNA binding domain (MBD) and NCoR-interaction domain (NID) of MeCP2 give rise to Rett syndrome[19] and by evidence that a mini-MeCP2 comprising only the MBD and NID rescues Rett-like neurological phenotypes in mice[17].

In mouse cells, MeCP2 is enriched within large foci of pericentric heterochromatin[1], also known as chromocenters[20], which contain heavily methylated satellite DNA repeats[21]. Recently, several studies have proposed that colocalization with heterochromatic foci is caused by an intrinsic tendency of MeCP2 to form 'condensates' via a process of liquid-liquid phase separation[22–26]. Distinguishing phase separation from alternative origins of intracellular 'membraneless compartments' in vivo is challenging[27–29]. In particular, whether MeCP2 forms phase-separated condensates or simply localises to regions enriched in DNA methylation is an open question.

In this study, we aim to determine the molecular determinants of MeCP2 foci in mouse cells using MeCP2 mutant constructs and genetically engineered cell lines lacking DNA methylation or organised

[1]The Wellcome Centre for Cell Biology, University of Edinburgh, Michael Swann Building, Max Born Crescent, The King's Buildings, Edinburgh EH9 3BF, UK. [2]The Roslin Institute and Royal (Dick) School of Veterinary Studies, University of Edinburgh, Easter Bush, Midlothian EH25 9RG, UK. [3]Max Planck Institute of Immunobiology and Epigenetics, Stübeweg 51, 79108 Freiburg, Germany. [4]Center for Inflammation Research, VIB, Ghent, Belgium. [5]Department of Biomedical Molecular Biology, Ghent University, Ghent, Belgium. ✉e-mail: a.bird@ed.ac.uk

heterochromatin. Furthermore, we investigate the influence of genome architecture on MeCP2 sub-nuclear distribution by performing live-cell imaging across 16 different mammalian species. Our results challenge the hypothesis that MeCP2 accumulation or heterochromatin formation requires phase separation in live cells and suggest that the prominent chromocenters observed in mouse cells are governed by atypical genomic features which are not present in most mammalian species.

## Results

### MeCP2 MBD is essential for localisation to DNA foci in mouse

The two best-characterised macromolecular interaction partners of MeCP2 are methylated DNA, which binds to the methyl-CpG binding domain (MBD)[30,31] and the NCoR co-repressor complex which binds to the NID[16] (see Fig. 1a). Previous work demonstrated that the MBD is essential for localisation to heavily methylated DNA-dense foci in mouse cells[32–34] and we confirmed this by transfection of mouse fibroblasts (NIH 3T3 cells) with constructs expressing EGFP-tagged mouse MeCP2 (Fig. 1a). In these and subsequent experiments we monitored the distribution of exogenous MeCP2 in living cells using high-resolution microscopy. Deletion of 27 amino acids within the MBD ($\Delta 99$–$125$)[32] greatly decreased co-localisation with DNA-dense foci, causing instead accumulation of MeCP2 in larger non-overlapping nuclear bodies (Fig. 1b). Therefore the two proposed intrinsically disordered regions (IDRs) of MeCP2 were unable to target MeCP2 to heterochromatin in the absence of a functional MBD. To test whether the MBD was sufficient for correct subnuclear localisation we expressed an EGFP-tagged 85 amino acid peptide corresponding to the minimal MBD[30]. Despite lacking the two IDRs of MeCP2 (IDR1 and IDR2), the minimal MBD localised to DNA-dense foci (Fig. 1b). To quantify this distribution, we compared the intensity of fluorescence within foci versus the remaining nucleoplasm (Supplementary Fig. 1a). This analysis confirmed the preference of the MBD for DNA-dense foci, suggesting that the MBD is sufficient as well as necessary for targeting heterochromatic foci. This conclusion is qualified by the observation that the level of focal fluorescence is significantly lower than that of full length MeCP2 (Fig. 1c). This may be due to the proximity of the relatively large EGFP tag, but it is also possible that other MeCP2 regions play a role in stabilising binding to chromocenters. For example, the minimal MBD lacks a nuclear localisation signal, which is not essential for MeCP2 entry into the nucleus, but may increase binding by raising its nuclear concentration[35]. Additionally, DNA binding specificity may be provided by three potential AT-Hooks[36,37], of which only AT-Hook1 shows a marked preference for AT-rich DNA in vitro[38] (Fig. 1a), although mutation of AT-Hook1 (R188G, R190G) revealed no detectable contribution to MeCP2 sub-nuclear localisation either in the presence or absence of a functional MBD (Fig. 1a–c). These findings in fibroblasts were replicated in mouse embryonic stem cells (ESCs) (Supplementary Fig. 1b, c). Taken together, the evidence indicates that the MBD is strictly necessary for heterochromatic localisation, but we cannot rule out that other regions of the protein, including perhaps 'intrinsically disordered regions', contribute to robust occupation of these sites.

### Heterochromatin is dispensable for MeCP2 binding in mouse

To determine the epigenomic requirements for MeCP2 distribution inside the nucleus, we first used ESCs lacking all three DNA methyltransferases: DNMT1, DNMT3A and DNMT3B[39]. These triple knockout (*DNMT TKO*) cells contain no detectable 5-methylcytosine, as confirmed by mass spectrometry of genomic DNA (Supplementary Fig. 2a). In agreement with a previous report[24], MeCP2 retained localisation to chromocenters in live *DNMT TKO* ESCs, although nucleoplasmic signal was also significantly elevated compared to the parental cell line (Fig. 1d, e). In fact, MeCP2 and Hoechst displayed closely similar patterns, implying generalised DNA binding in the absence of

DNA methylation. Inactivation of AT-Hook 1 had no effect on this nuclear distribution. Neither the MBD alone, nor full-length MeCP2 lacking a functional MBD, targeted chromocenters in *DNMT TKO* ESCs (Supplementary Fig. 2b, c), indicating that the full-length protein is required for heterochromatic localisation in the absence of DNA methylation. These findings raised the possibility that MeCP2 localisation in the absence of DNA methylation is relatively non-specific, in which case the dynamics of chromatin binding would be expected to dramatically increase. To test this prediction, we performed fluorescence recovery after photobleaching (FRAP) of wild-type MeCP2 at heterochromatic foci. In line with previous studies[9,34,36,40], wild-type ESCs showed incomplete fluorescence recovery, even >6 min after bleaching, whereas MeCP2 recovery was complete and rapid in *DNMT TKO* ESCs (Fig. 1f, Supplementary Fig. 2d). Incomplete recovery and failure of fluorescence to plateau in wild-type cells prevents simple numerical comparison between wild-type and mutant cells, but it is evident that the time to reach 50% recovery is greatly reduced in *DNMT TKO* cells (-10 s versus ~75 s; Table 1). The data show that the stably bound fraction of MeCP2 is abolished in the absence of DNA methylation and MeCP2 binding becomes much more transient and dynamic, suggesting a loss of DNA binding specificity.

The DNA-dense foci in mouse cells at which MeCP2 accumulates correspond to pericentric heterochromatin, which is marked by trimethylation of histone H3 at lysine 9 (H3K9me3). To evaluate whether MeCP2 co-localisation depends on intact heterochromatin organisation, we used mouse cell lines in which histone lysine methyltransferase genes have been deleted[41]. *Suv39h1/2* double knock-out (*2KO*) fibroblasts lack H3K9 trimethylation, whereas in *5KO* fibroblasts, five out of six known H3K9 methyltransferases were deleted (Suv39h1/2, Eset2, and G9a/Glp) as *Eset1* knockout is lethal (Supplementary Fig. 3a, b)[41]. Interestingly, *5KO* cells lack all canonical heterochromatin marks H3K9me1/2/3 (Supplementary Fig. 3c). Of note, DNA methylation is largely unaffected in these cells[41]. Immunostaining of fixed cells confirmed loss of H3K9me3 at DNA-dense foci and diffuse localisation of heterochromatin protein 1α (HP1α), both in *2KO* and *5KO* fibroblasts (Supplementary Fig. 3d, e). To visualise heterochromatin in live cells, we co-transfected our cell lines with a reporter that expressed a dimerised HP1β (CBX1) chromodomain fused to a fluorescent protein (CHD-mCherry) (Fig. 2a)[42]. As expected, CHD-mCherry signal became diffuse in *2KO* and *5KO* cell lines (Fig. 2b). Strikingly, the nuclear distribution of full-length MeCP2 appeared unaltered by the dissolution of heterochromatin in these cells (Fig. 2b, c). Similar to our results in wild-type cells, the MBD is critical for proper localisation of MeCP2 to chromocenters in *5KO*, but the minimal MBD alone showed reduced binding (Supplementary Fig. 4a, b). Furthermore, FRAP analysis showed no difference in fluorescence recovery between the *5KO* and wild-type cell lines (Fig. 2d, Supplementary Fig. 4c, d and Table 1), suggesting identical dynamics of MeCP2 binding whether heterochromatin is intact or dispersed. To test whether loss of MeCP2 has a direct impact on pericentric heterochromatin organisation, we performed immunofluorescence analysis in *Mecp2* knockout fibroblasts[10] (Supplementary Fig. 5a). H3K9me3-positive (Supplementary Fig. 5b) and HP1α-positive (Supplementary Fig. 5c) foci co-localising with DNA were indistinguishable from the wild-type control. The results suggest that, despite their co-location, MeCP2 accumulation and heterochromatin organisation are mutually independent phenomena.

Transfection experiments allowed us to test further predictions of the hypothesis that MeCP2 modulates the structure of heterochromatic foci via its involvement in liquid-liquid phase separation[22,26,43,44]. This model predicts that increasing MeCP2 expression will result in larger chromocenters while maintaining a fixed concentration of MeCP2 ($c_{sat}$) within each focus (Fig. 2e); a property defined as 'concentration buffering'[45]. To test this, we analysed transfected *Mecp2*-null cells expressing varying levels of MeCP2 and found that the concentration of MeCP2 within foci did not remain constant but

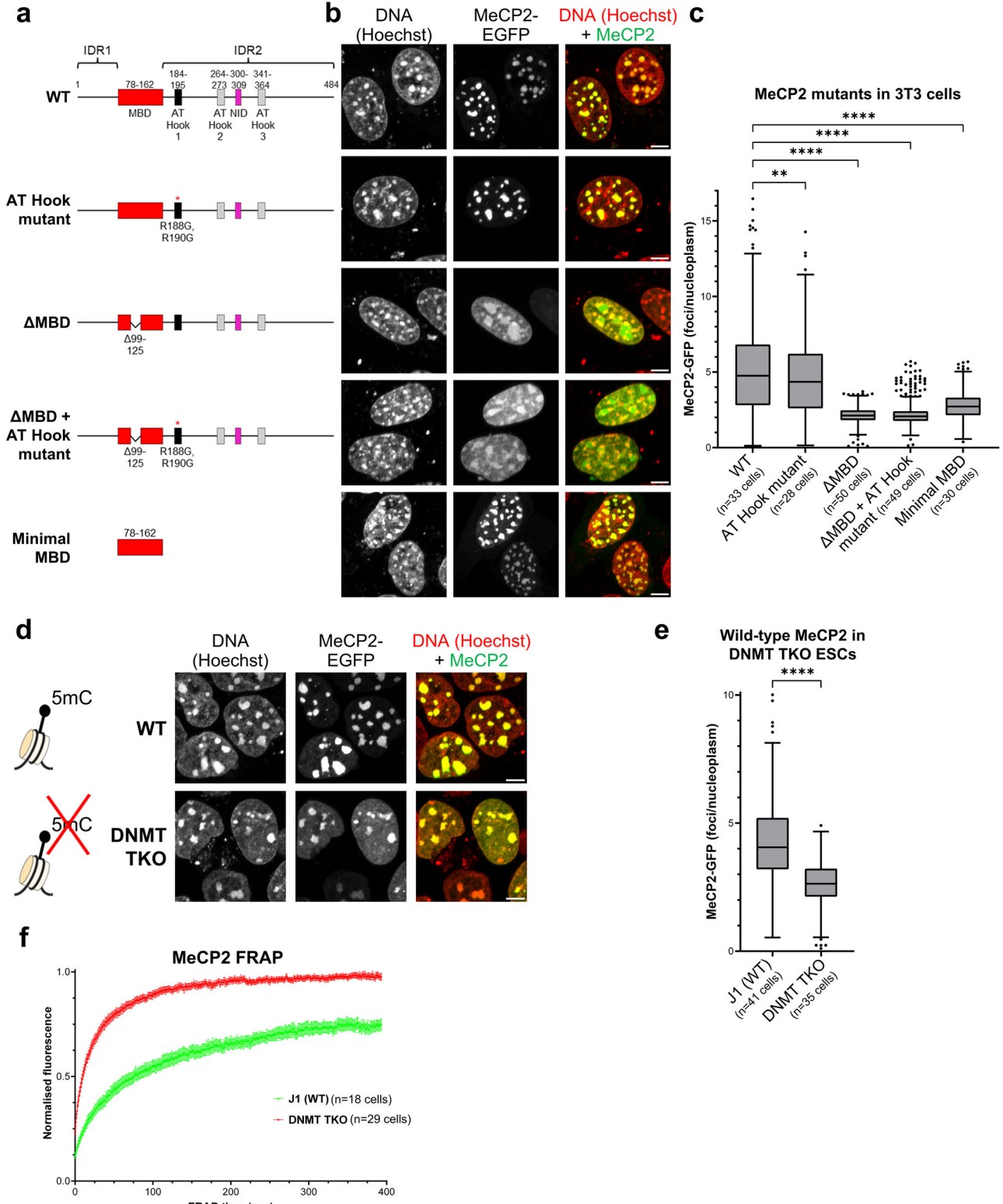

correlated with total MeCP2 expression levels (Fig. 2f, g, Supplementary Fig. 5d). As expected from a 'size buffering' model (Fig. 2e), the increase in MeCP2 within foci was also accompanied by increased MeCP2 concentration within the nucleoplasm, resulting in a relatively stable foci/nucleoplasm ratio of fluorescence signal (Supplementary Fig. 5d). Although we observed a small difference in the average size of chromocenters between the groups of low- and high-MeCP2 expressing cells (Supplementary Fig. 5d), chromocenter size correlated very poorly with MeCP2 expression levels (Fig. 2g). Additionally, the

number of heterochromatic foci per cell was not influenced by changes in MeCP2 expression (Fig. 2g, Supplementary Fig. 5d). Overall, these observations fail to support the phase separation model.

## MeCP2 distribution is diffuse in most mammalian species
A prediction of the phase separation hypothesis is that MeCP2, whose amino acid sequence is highly conserved in vertebrates, will form equivalent nuclear foci in cells from diverse mammalian species. To test this, we obtained cell lines (mainly primary fibroblasts) from 16

**Fig. 1 | MeCP2 localises to DNA-dense foci in mouse cells via its Methyl-DNA Binding Domain (MBD). a** Diagram showing the structure of wild-type and mutant MeCP2 constructs (fused with EGFP) used for live-cell imaging with annotated domains. MBD methyl-CpG binding domain, NID NCoR-interaction domain, IDR intrinsically disordered region. **b** Live-cell imaging of 3T3 cells transfected with the EGFP-MeCP2 constructs indicated in (**a**). Hoechst staining was used to visualise DNA. Scale bars: 5 μm. **c** Box plot showing the quantification of MeCP2 wild-type and mutant fluorescence at DNA-dense foci (relative to nucleoplasm) in 3T3 cells, as described in (**b**). The box lower and upper limits correspond to the 25th and 75th percentiles, respectively, with the centre line corresponding to the median. Whiskers extend up to 1.5 times the interquartile distance according to Tukey's method, and individual points are outliers. The number of analysed cells from two independent experiments are: WT $n = 33$ cells, AT-Hook mutant $n = 28$ cells, ΔMBD $n = 50$ cells, ΔMBD + AT-Hook mutant $n = 49$ cells, Minimal MBD $n = 30$ cells. Stars indicate statistical significance compared to wild-type MeCP2 (Brown-Forsythe and

Welch ANOVA test). **d** Live-cell imaging of wild-type (J1) and *DNMT TKO* ESCs transfected with wild-type EGFP-MeCP2. Hoechst staining was used to visualise DNA. Scale bars: 5 μm. Diagram adapted from Lyst and Bird, Nat Rev Genet, 2015[19]. **e** Box plot showing the quantification of MeCP2 fluorescence at DNA-dense foci (relative to nucleoplasm) in wild-type (J1) and *DNMT TKO* ESCs, as described in (**d**). The box lower and upper limits correspond to the 25th and 75th percentiles, respectively, with the centre line corresponding to the median. Whiskers extend up to 1.5 times the interquartile distance according to Tukey's method, and individual points are outliers. The number of analysed cells from two independent experiments are: J1 (WT) $n = 41$ cells, *DNMT TKO* $n = 35$ cells. Stars indicate statistical significance compared to wild-type cells (two-tailed unpaired *t* test with Welch's correction). **f** Graph showing the FRAP quantification of wild-type EGFP-MeCP2 in wild-type (J1, green) and *DNMT TKO* (red) ESCs. The number of analysed cells from two independent experiments are: J1 (WT) $n = 18$ cells, *DNMT TKO* $n = 29$ cells. Error bars: SEM.

**Table 1 | MeCP2 FRAP analysis in mutant cell lines**

| Sample | Mobile fraction (plateau) | Immobile fraction | $T_{50}$ (50% recovery) |
|---|---|---|---|
| WT J1 ESCs ($n = 18$ cells) | 75.2% ± 4.7% | 24.8% ± 4.7% | 75.5 s ± 13 s |
| DNMT TKO ESCs ($n = 29$ cells) | 97.5% ± 1.6% | 2.5% ± 1.6% | 9.9 s ± 0.3 s |
| WT Eset25 fibroblasts ($n = 28$ cells) | 74.8% ± 1.3% | 25.2% ± 1.3% | 105 s ± 28 s |
| 5KO fibroblasts ($n = 26$ cells) | 77.5% ± 5.6% | 22.5% ± 5.6% | 84.2 s ± 30 s |

different species covering most mammalian lineages (Fig. 3a, Supplementary Data 1) and transfected each with MeCP2-EGFP and CHD-mCherry. Surprisingly, high-resolution microscopy revealed a uniformly diffuse nuclear distribution of MeCP2 in the great majority of cell lines, including humans. Furthermore, localisation of the heterochromatin reporter was dispersed throughout the nucleus, and chromocenters, characterised by intense foci of Hoechst staining in live cells, were absent in most mammalian species (Fig. 3b, c and Supplementary Fig. 6). The only exceptions were mouse and red deer cells, where MeCP2 and heterochromatin co-localised within prominent DNA foci. For unknown reasons, foci in red deer cells were only detected in about half of nuclei, the remainder showing a diffuse distribution. In addition, a rare subpopulation of cow and monkey cells (5–10% of the cells) showed spotty MeCP2 signal which did not colocalise with foci of heterochromatin or DNA (Supplementary Fig. 7a). The results indicate that MeCP2 does not have an intrinsic tendency to adopt a focal organisation.

MeCP2 of mice was expressed in all transfection experiments as the MeCP2 protein sequence is highly conserved across mammals (Supplementary Fig. 8). Notably, all residues within the MBD are identical in all the species analysed in this study. To test whether the observed nuclear distributions could nevertheless be due to the non-homologous origin of MeCP2, we transfected human MeCP2 into mouse and human fibroblasts. Again, prominent nuclear foci were observed in mouse cells, but MeCP2 was dispersed in human nuclei (Supplementary Fig. 7b). We also asked whether endogenous MeCP2 matched the distribution seen in transfection experiments. Although MeCP2 levels are low in fibroblasts, we could verify in several lines by immunostaining that the endogenous MeCP2 pattern (Supplementary Fig. 7c) is similar to that of transfected mouse MeCP2 (Fig. 3b and Supplementary Fig. 6). Since MeCP2 is most abundant and functionally important in the nervous system, we also asked whether MeCP2 adopts a distinct distribution in neurons. Using human dopaminergic neurons derived from Lund Human Mesencephalic (LUHMES) cells[46,47] that express endogenously tagged MeCP2-mCherry protein[48] (Supplementary Fig. 9a), we confirmed by live-cell imaging a diffuse nuclear pattern of MeCP2 (Fig. 3d). Additionally, we performed MeCP2 immunostaining in mouse and rat brain sections. In striking contrast to mice, MeCP2 signal in rat neurons

was mainly diffuse both in the cortex (Fig. 3e) and hippocampus (Supplementary Fig. 9b), both of which are physiologically relevant regions for MeCP2 function. These findings indicate that the dramatic differences in MeCP2 nuclear localisation among mammals are not due to interspecific variation in their MeCP2 proteins or to the tissue origin of the cell type under investigation but are determined by the prevailing genome structure in each species.

DNA methylation patterns are a prime candidate for determining whether MeCP2 is dispersed or concentrated in the nucleus (Fig. 3c). Accordingly, immunostaining with an antibody directed against 5-methylcytosine showed large clusters resembling MeCP2 foci in mouse and red deer cells, while other species displayed diffuse patterns (Supplementary Fig. 10a). This observation is in agreement with the well-documented accumulation of methylated major satellite DNA at pericentric heterochromatin in mouse cells[49-55]. Like mice, the red deer genome contains highly repetitive elements, including satellite I DNA which localises to pericentromeric regions[56-58]. To test whether red deer satellite repeat elements are methylated, we performed a Southern blot of genomic DNA digested with methylation-sensitive (HpaII) or methylation-insensitive (MspI) isoschizomers (Supplementary Fig. 10b). Probing with the satellite I DNA sequence confirmed that the repeat arrays are heavily methylated (Supplementary Fig. 10c). Agarose gel electrophoresis also confirmed that satellite I DNA repeats are highly abundant, as they were visible as discrete bands in bulk genomic DNA by ethidium bromide staining. The results suggest that the demarcation of prominent chromocenters is dependent on reiteration of underlying satellite repeat DNA arrays and the presence of DNA methylation within the repeated motifs is the key genomic feature that leads to MeCP2 foci.

To further explore this hypothesis, we compared fibroblast cell lines from two closely related mouse species that have dramatically different amounts of satellite repeat DNA: *Mus musculus*, the most widely used mouse model, and *Mus spretus* which diverged from *M. musculus* 1–2 million years ago[59,60]. In contrast to *Mus musculus* cells, the *Mus spretus* genome has barely detectable levels of major satellite DNA at pericentric regions[61,62] (Fig. 4a), as confirmed by fluorescence in situ hybridisation (Fig. 4b) and Southern blot analysis (Fig. 4c). Most major satellite repeats contain cleavage sites for both ApoI and Hpy-CH4IV (Supplementary Fig. 10d), but the latter enzyme cuts rarely due

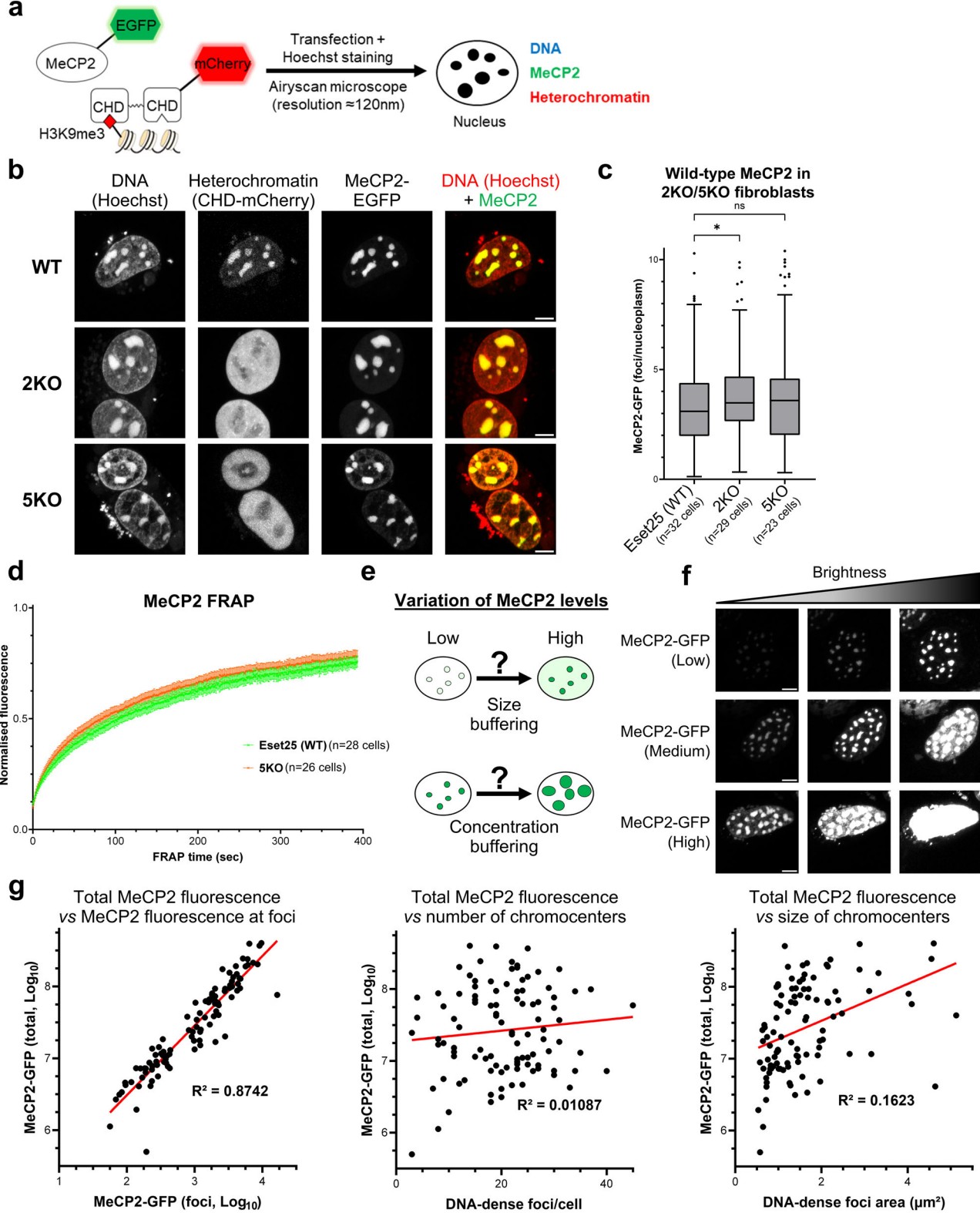

to methylation within its ACGT recognition sequence. Accordingly, immunostaining showed that 5-methylcytosine is highly enriched at *Mus musculus* nuclear foci but appeared relatively diffuse in *Mus spretus* cells (Fig. 4d). Immunofluorescence and live-cell imaging confirmed the absence of DNA-dense foci in *Mus spretus*, as well as diffuse patterns of MeCP2 and heterochromatic markers (Fig. 4b, e, f). Overall, we conclude that the two exceptional species in the panel of mammals, mouse and red deer, differ from the majority by having satellite repeat

DNA sequences near centromeres that are both highly abundant and heavily methylated (Fig. 4g).

## Discussion

The goal of this study was to unravel the relationship between MeCP2 accumulation, DNA methylation and heterochromatin organisation. In particular, we wished to assess using live-cell imaging the validity of recent claims that MeCP2 contributes to heterochromatin formation

**Fig. 2 | Heterochromatin organisation does not determine MeCP2 localisation. a** Diagram adapted from Lyst and Bird, Nat Rev Genet, 2015[19] showing the strategy used to visualise MeCP2 (EGFP fusion protein), heterochromatin (HP1 chromodomain reporter) and DNA (Hoechst staining) in live cells by high-resolution confocal microscopy. **b** Live-cell imaging of wild-type (Eset25) and H3K9 lysine methyltransferases knockout fibroblasts (2KO/5KO) transfected with wild-type EGFP-MeCP2. Hoechst staining and a CHD-mCherry reporter were used to visualise DNA and heterochromatin, respectively. Scale bars: 5 μm. **c** Box plot showing the quantification of MeCP2 fluorescence at DNA-dense foci (relative to nucleoplasm) in wild-type (Eset25), 2KO and 5KO fibroblasts, as described in (**b**). The box lower and upper limits correspond to the 25th and 75th percentiles, respectively, with the centre line corresponding to the median. Whiskers extend up to 1.5 times the interquartile distance according to Tukey's method, and individual points are outliers. The number of analysed cells from three independent experiments are: Eset25 (WT) $n = 32$ cells, 2KO $n = 29$ cells, 5KO $n = 23$ cells. Stars indicate statistical significance compared to wild-type cells (Brown–Forsythe and Welch ANOVA test).

**d** Graph showing the FRAP quantification of wild-type EGFP-MeCP2 in wild-type (Eset25) and 5KO fibroblasts. The number of analysed cells from two independent experiments are: Eset25 (WT) $n = 28$ cells, 5KO $n = 26$ cells. Error bars: SEM. **e** Diagram showing two possible responses of MeCP2-containing foci to variations in expression levels. **f** Live-cell imaging of Mecp2 knockout fibroblasts transfected with varying levels of wild-type EGFP-MeCP2. Cells were divided into three expression categories (low, medium, high) and images are shown at three levels of brightness to enable comparison. Scale bars: 5 μm. **g** Scatterplots showing the relationship between total MeCP2 expression levels within cells (as described in (**f**)) and different parameters associated with MeCP2 foci. Values were plotted for individual cells ($n = 101$ cells) from two independent experiments. MeCP2 fluorescence (total and within foci) was expressed on a logarithmic scale to account for the broad distribution of the data. The calculated $R^2$ values indicate a significant correlation only between total MeCP2 expression and MeCP2 concentration within foci.

by undergoing phase separation. We found that the prominent MeCP2-containing foci seen in mouse cells are absent in 14 out of 16 mammalian species. Typically, the nuclear distribution of MeCP2 is diffuse in mammalian cells, including human neurons, indicating that MeCP2 does not intrinsically form large condensates. While purified MeCP2 protein forms liquid droplets in vitro, it fails to display key hallmarks of liquid–liquid phase separation in vivo[45,63]. Notably, in mouse cells we found no evidence for a critical concentration ($c_{sat}$) at which MeCP2 would form condensates and showed that the size of MeCP2-containing foci is uncoupled from total MeCP2 expression levels. Our conclusions agree with a previous study that used live imaging approaches (including the half-bleach FRAP assay) to show that the proteins HP1 and MeCP2 do not behave as members of a phase-separated compartment in mouse cells[64].

The results also question the notion that mouse pericentric heterochromatin depends upon phase separation, as a heterochromatin reporter[42] that recognises the histone mark H3K9me3 via its chromodomains revealed a diffuse nuclear pattern in most mammalian species. While heterochromatin is usually thought to involve chromatin compaction, most mammals in fact show little evidence of H3K9me3 clustering. These findings cast doubt on the proposal that heterochromatin-associated proteins, including HP1, drive heterochromatin condensation[65,66]. As previously proposed, heterochromatic foci in mouse cells may resemble 'collapsed chromatin globules' rather than phase-separated condensates[64]. The coalescence of pericentric repetitive DNA may result from structural features of satellite sequences, perhaps involving DNA motifs that promote alternative conformations[67]. The AT-rich base composition of mouse major satellite is probably not required for condensation as satellite I of red deer, which also forms chromocenters, is markedly more GC-rich than bulk genomic DNA (55% versus ~40% GC).

Our data are compatible with a simple biochemical explanation for nuclear localisation of MeCP2 based on its affinity for methylated DNA[28,32–34,68–70]. The 85 amino acid MBD of MeCP2 alone is sufficient to target DNA-dense foci in mouse cells, although with lower efficiency compared to the full-length protein. Interestingly, the minimal MBD localises to chromocenters only in the presence of DNA methylation, while mutation of the MBD within the full-length protein abolishes MeCP2 subnuclear localisation. Although the AT-Hooks of MeCP2 may contribute to MeCP2 binding dynamics via weak and transient interactions with genomic DNA[70], these motifs are dispensable for MeCP2 localisation to heterochromatic foci. Moreover, mutations that inactivate the AT-hook in humans apparently do not cause Rett syndrome[38]. Similarly, with the exception of the NID, IDRs are largely dispensable for MeCP2 function, as transgenic mice expressing a truncation of most N- and C-terminal IDRs show no overt phenotype[17]. As previously noted[24], MeCP2 retains its localisation to DNA-dense foci in mouse cells in the complete absence of

DNA methylation (DNMT TKO ESCs), and this phenotype relies on an intact MBD domain. This could be explained by methylation-independent functions of the MBD, such as a weak intrinsic affinity for AT-rich sequences[36], or protein-protein interactions with heterochromatin-associated proteins such as ATRX[71]. However, the exchange rate of MeCP2 in DNMT TKO ESCs as measured by FRAP is greatly increased compared to wild-type cells, indicating reduced DNA binding affinity. It is notable that mutations that similarly reduce the residence time of MeCP2 on mouse heterochromatin cause Rett syndrome[69], suggesting that fast exchange is incompatible with MeCP2 function.

The nuclear distribution of MeCP2 appears to be directly influenced by variations in genomic DNA methylation patterns between species. In Mus musculus and red deer, MeCP2 coincides with dense clusters of 5-methylcytosine at arrays of satellite repeat sequences, whereas in the other 14 mammalian species tested, both DNA methylation and MeCP2 are relatively uniformly distributed, consistent with the global distribution of methylated CGs[72–75]. The importance of abundant repetitive DNA elements for the creation of MeCP2 foci is illustrated by the differences between two mouse species that are sufficiently closely related to form hybrids: Mus musculus and Mus spretus. Mus spretus cells lack most major satellite DNA repeats[61,62], have no apparent chromocenters and, in striking contrast to the 'classic' mouse model (Mus musculus), display diffuse nuclear MeCP2 and DNA methylation patterns. We note that methylation of sequences other than CG, in particular mCA which is abundant in mature mammalian neurons and serves as an additional target for MeCP2 binding, is probably very rare in the non-neuronal cells and in vitro differentiated neurons studied here[2,7,76]. There is no reason to expect that binding to mCA versus mCG would lead to differential higher-order accumulation of the MeCP2 protein[77]. Accordingly, we found that MeCP2 distribution is also diffuse in cortical and hippocampal neurons of rat brains, where mCA is present.

Although MeCP2 localises prominently to heterochromatic foci in mice, the protein is also bound genome wide to euchromatin, where mCG sites (and mCA in neurons) are highly abundant. While the relationship between MeCP2 binding to euchromatic genes and transcriptional regulation has been extensively characterised[3–9], the functional significance of MeCP2 at pericentric heterochromatin remains unclear. Previous studies using cellular models proposed a role for MeCP2 in promoting the clustering of chromocenters[40,43,44,68] or by controlling the partitioning of HP1(α/γ) within foci[22,78]. Additionally, it was reported that MeCP2 directly recruits H3K9 methyltransferase activity[79]. However, MeCP2-null neurons in mice show patterns of H3K9me3 and HP1 in heterochromatin which are largely indistinguishable from wild-type, although slightly increased DAPI staining within foci and altered patterns of ATRX or H4K20me3 have

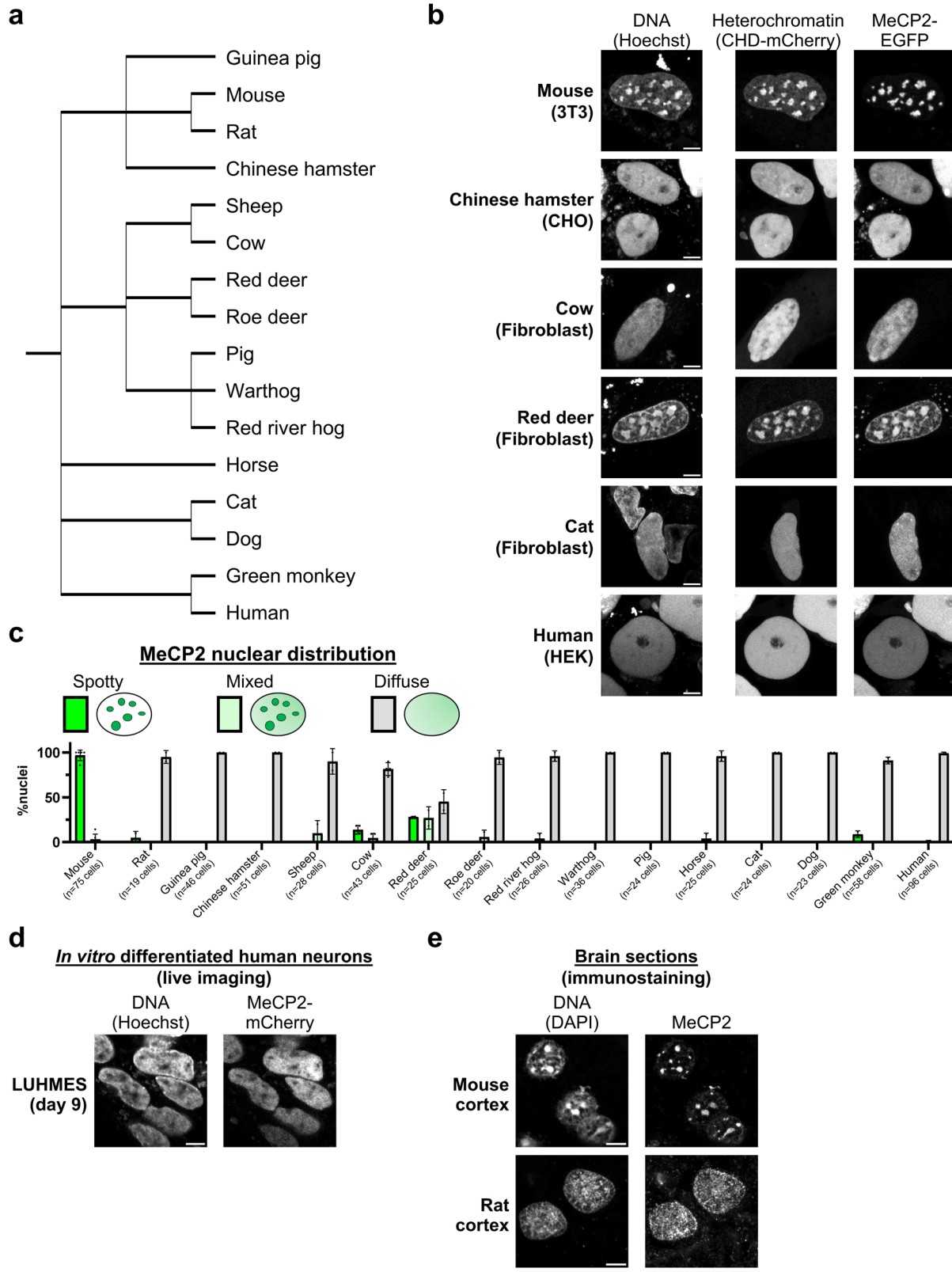

**c** MeCP2 nuclear distribution

Spotty · Mixed · Diffuse

**d** *In vitro* differentiated human neurons (live imaging)

DNA (Hoechst) · MeCP2-mCherry

LUHMES (day 9)

**e** Brain sections (immunostaining)

DNA (DAPI) · MeCP2

Mouse cortex

Rat cortex

been reported[71,80,81]. Moreover, complete loss of H3K9 methylation and the consequent dispersion of HP1α does not affect MeCP2 localisation to foci in mouse cells and has no impact on its DNA binding dynamics as assessed by FRAP. Thus, taken together, the results presented here argue that the biogenesis of heterochromatin compartments does not involve or require MeCP2.

## Methods

### Cell culture

A detailed list of all cell lines used in this study is provided as a supplementary datasheet (see Supplementary Data 1). All cell lines were grown in incubators at 37 °C with 5% $CO_2$. Fibroblasts and immortalised cell lines were cultured in Glasgow minimum essential medium (GMEM; Gibco ref. 11710035) supplemented with 10% foetal bovine

**Fig. 3 | MeCP2 displays a diffuse nuclear distribution in most mammalian species. a** Phylogenetic tree showing the mammalian species used in this study (from NCBI Taxonomy). **b** Live-cell imaging of the indicated mammalian cell lines transfected with wild-type EGFP-MeCP2. Hoechst staining and a CHD-mCherry reporter were used to visualise DNA and heterochromatin, respectively. Scale bars: 5 μm. **c** Graph showing quantification of MeCP2 nuclear distribution (spotty, mixed, diffuse) in all studied mammalian species, as described in (**b**) and Supplementary Fig. 6. The number of analysed cells from at least two independent experiments are: Mouse $n = 75$ cells, Rat $n = 19$ cells, Guinea pig $n = 46$ cells, Chinese hamster $n = 51$ cells, Sheep $n = 28$ cells, Cow $n = 43$ cells, Red deer $n = 25$ cells, Roe deer $n = 20$ cells, Red river hog $n = 26$ cells, Warthog $n = 36$ cells, Pig $n = 24$ cells, Horse $n = 25$ cells, Cat $n = 24$ cells, Dog $n = 23$ cells, Green monkey $n = 58$ cells, Human $n = 96$ cells. Data are presented as mean values ± SD. **d** Live-cell imaging of human postmitotic neurons (LUHMES cells) expressing endogenously tagged MeCP2-mCherry (representative images from two independent experiments). Hoechst staining was used to visualise DNA. Scale bar: 5 μm. **e** Immunofluorescence of endogenous MeCP2 in mouse and rat brain cortex (representative images from multiple sections in a single animal). DAPI staining was used to visualise DNA. Scale bars: 5 μm.

serum (batch tested), 1x L-glutamine (Gibco ref. 25030024), 1x MEM non-essential amino acids (Gibco ref. 11140035), 1 mM sodium pyruvate (Gibco ref. 11360039), 0.1 mM 2-mercaptoethanol (Gibco ref. 31350010). Eset25, *2KO* and *5KO* cell lines were grown in Dulbecco's Modified Eagle Medium (DMEM, Gibco ref. 41966029) supplemented with 10% foetal bovine serum and 1.25 μg/ml puromycin. ESC lines were cultured in gelatin-coated dishes containing GMEM supplemented with 10% foetal bovine serum, 1x L-glutamine, 1x MEM non-essential amino acids, 1 mM sodium pyruvate, 0.1 mM 2-mercaptoethanol and 100 U/ml leukaemia inhibitory factor (LIF, batch tested). LUHMES cells were cultured and differentiated as described in ref. [47,48] in poly-L-ornithine and fibronectin-coated dishes. Primary cell lines were derived from tissue explants or collagenase dissociated tissues and expanded in GMEM supplemented with 10% foetal bovine serum, 2 mM L-glutamine, 1x MEM non-essential amino acids, 1 mM sodium pyruvate, 0.1 mM 2-mercaptoethanol, 100 U/ml Penicillin-Streptomycin, 50 μg/ml Gentamicin and 2.5 μg/ml Amphotericin B. Once established, cultures were maintained in the same medium without antibiotics. *Mus spretus* primary fibroblasts were immortalised using SV40 virus as described in ref. [82]. For imaging, cells were plated and cultured directly on polymer coverslips (iBidi cat. 81156 or 80286) with the appropriate coating and culture conditions.

As a quality control, all mammalian cell lines were tested for Mycoplasma contamination (Lonza cat. LT07-218). Additionally, the identity of each cell line was verified by Sanger sequencing of mitochondrial Cytochrome b which was amplified from extracted total DNA using universal primers (FW: CGAAGCTTGATATGAAAAACCATCGTTG, RV: AAACTGCAGCCCCTCAGAATGATATTTGTCCTCA)[83,84].

### Animal models (mouse and rat)
All experiments involving mice and rats were approved by the local Animal Welfare Ethical Review Body (AWERB) of the University of Edinburgh and were part of project licences approved by the UK Home Office (PP4326006 and PP4366223) and in accordance with the Animal and Scientific Procedures Act 1986. The rodents used in this study were bred and maintained at the University of Edinburgh animal facilities under standard conditions with 12 h dark/light cycles, an ambient temperature of 20–24 °C and relative humidity of 45–65%. All procedures were carried out by staff licensed by the UK Home Office and in accordance with the Animal and Scientific Procedures Act 1986. Brain tissue was harvested from one 4-week-old Long-Evans hooded wild-type male rat, and from one 10-week-old wild-type male mouse on a mixed C57BL/6 × CBA background. Tissue was directly snap frozen in liquid nitrogen or on dry ice and stored at −80 °C. Both rodents used were male. Collected tissues were used to determine MeCP2 nuclear distribution by immunofluorescence, which is not influenced by the sex animals. Therefore, sex was not considered as a key parameter in the design of our study.

### Molecular cloning
For the heterochromatin reporter, the dimerised CBX1 chromodomain from a previously described engineered chromatin reader[42] was subcloned into a pPYCAG mammalian expression vector[85] containing mCherry. MeCP2 mutant constructs were cloned by Gibson assembly (NEB cat. E5520S) using synthetic double-stranded DNA fragments ordered from Integrated DNA Technologies (IDT). All plasmids used in this study are detailed in Table 2.

### Live-cell imaging
Cells were transfected with MeCP2-EGFP with or without CHD-mCherry plasmid using Lipofectamine 3000 (Thermo Fisher Scientific cat. L3000008) and following the manufacturer's protocol. Cells were analysed the next day following a medium change and Hoechst 33342 (Thermo Fisher Scientific cat. R37605) staining using the Zeiss LSM 880 confocal microscope (with Airyscan module) at 37 °C with 5% $CO_2$. Images were acquired using the Zeiss ZEN software (black edition) and processed using the software Fiji (based on ImageJ v1.53[86]).

To quantify fluorescence at DNA-dense foci in mouse cells, images were processed using a custom script (https://doi.org/10.5281/zenodo.7740611). The detection of nuclei was automated by performing image segmentation on the Hoechst fluorescence channel (maximum intensity Z-projection) using the software Cellpose[87] integrated into the script using the BIOP ijl-utilities-wrappers plugin (v0.3.19). DNA-dense foci (chromocenters) were detected by applying thresholds per nucleus and several parameters were measured including the area of foci and nuclei ($μm^2$), fluorescence levels (average pixel intensity) of MeCP2/Hoechst at foci and within the nucleoplasm (total nuclear signal depleted from foci). For statistical comparisons between two conditions, we performed a two-tailed unpaired $t$ test with Welch's correction using the software GraphPad Prism v9.5.1. For statistical comparisons between three or more conditions, we performed a Brown–Forsythe and Welch one-way ANOVA test using the Games-Howell's method to correct for multiple comparisons (adjusted $p$ value) using the software GraphPad Prism v9.5.1. Statistical significance was defined by a $p$ value $\leq 0.05$ (ns $p > 0.05$, *$p \leq 0.05$, **$p \leq 0.01$, ***$p \leq 0.001$, ****$p \leq 0.0001$). For box plots, outliers were defined following Tukey's method as data points ranging beyond 1.5 times the interquartile distance (the difference between the 25th and 75th percentiles). For the quantification of MeCP2 nuclear distribution in mammalian species, all transfected cells were counted and classified into three categories (spotty, mixed and diffuse). Results were plotted using GraphPad Prism v9.5.1.

### Fluorescence recovery after photobleaching (FRAP)
Cells were transfected with wild-type MeCP2-EGFP using Lipofectamine 3000 (Thermo Fisher Scientific cat. L3000008) and following manufacturer's protocol. Cells were analysed the next day following a medium change, and FRAP was performed as previously described[9] using the Zeiss LSM 880 confocal microscope at 37 °C with 5% $CO_2$. For each cell, EGFP-MeCP2 signal was imaged every 1 s for 400 s with five images recorded before bleaching a selected MeCP2 spot (FRAP spot) with 100% laser power. Images were acquired using the Zeiss ZEN software (black edition) and processed using the software Fiji.

FRAP analysis of two independent transfection experiments was performed as previously described in ref. [9] using a custom macro with Fiji software (https://doi.org/10.5281/zenodo.2654601).

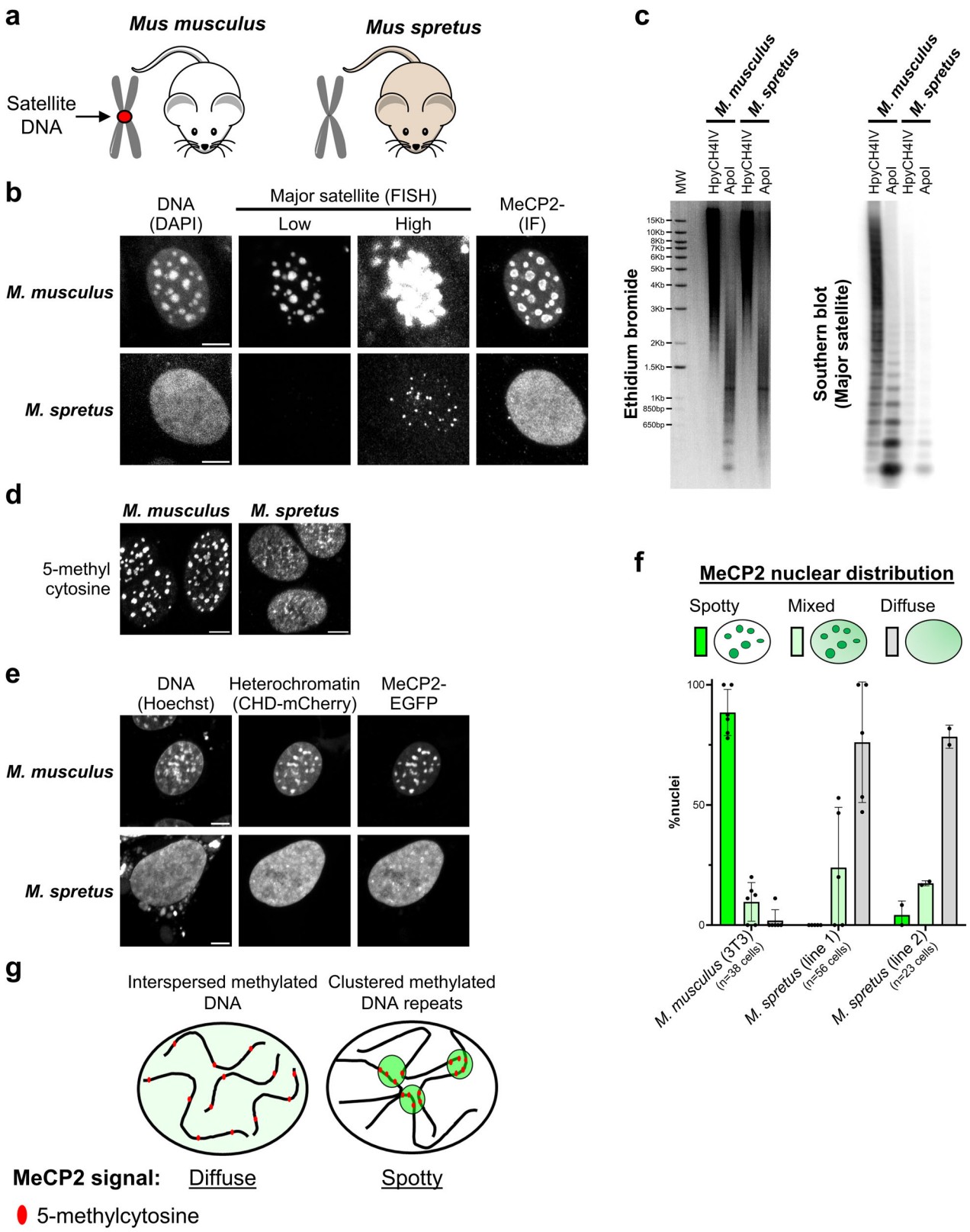

Fluorescence was measured at the bleached MeCP2 spot (FRAP spot), as well as a non-bleached MeCP2 spot (control spot) to account for photobleaching during the experiment. Additionally, the fluorescence outside of transfected cells was measured as background. The first time point ($T_0$) was defined as the first post-bleach image. For each time point, the FRAP fluorescence signal was normalised to fluorescence values before photobleaching, as described in the equation below:

$$Normalised\ FRAP = \frac{(FRAP\ spot)t - (Background)t}{(control\ spot)t - (Background)t}$$
$$\times \frac{(control\ spot)pre - bleach - (Background)pre - bleach}{(FRAP\ spot)pre - bleach - (Background)pre - bleach}$$

**Fig. 4 | MeCP2 foci depend on the presence of abundant satellite DNA repeats. a** Diagram showing two closely related mouse strains used in this study. The *Mus musculus* genome contains abundant major satellite DNA repeats which cluster at pericentromeric regions, while *Mus spretus* lacks these repetitive elements. **b** Fluorescence in situ hybridisation of major satellite DNA (shown at two levels of brightness) combined with immunofluorescence of MeCP2 in *Mus musculus* (3T3) and *Mus spretus* cell lines (representative images from two independent experiments). DAPI staining was used to visualise DNA. Scale bars: 5 μm. FISH Fluorescence in situ hybridisation, IF Immunofluorescence. **c** Ethidium bromide staining (left) and Southern blot (right) using a probe for major satellite DNA with *Mus musculus* and *Mus spretus* genomic DNA digested with a methylation-sensitive (HpyCH4IV) or -insensitive (ApoI) restriction enzyme (representative images from two independent experiments). MW molecular weight marker. **d** Immunofluorescence of antibody stained 5-methylcytosine in *Mus musculus* (3T3)

and *Mus spretus* cell lines (representative images from two independent experiments). Scale bars: 5 μm. **e** Live-cell imaging of *Mus musculus* (3T3) and *Mus spretus* cell lines transfected with wild-type EGFP-MeCP2. Hoechst staining and a CHD-mCherry reporter were used to visualise DNA and heterochromatin, respectively. Scale bars: 5 μm. **f** Graph showing the quantification of MeCP2 nuclear distribution (spotty, mixed, diffuse) in *Mus musculus* (3T3) and *Mus spretus* cell lines, as described in (**e**). The number of analysed cells from at least two independent experiments are: *M. musculus* (3T3) $n = 38$ cells, *M. spretus* (line 1) $n = 56$ cells, *M. spretus* (line 2) $n = 23$ cells. Data are presented as mean values ± SD. **g** Model for differential MeCP2 nuclear distribution in mammalian species. In most species, including humans, methylated DNA elements are interspersed along the genome leading to a diffuse MeCP2 nuclear pattern. In some cases, like *Mus musculus*, methylated DNA repeats cluster into 'chromocenters' leading to spotty MeCP2 signal.

## Table 2 | List of plasmids used in the study

| Mammalian expression plasmids | | |
|---|---|---|
| **Name** | **Details** | **Source** |
| pPYCAG-mCherry-2xCHD | Heterochromatin reporter (dimerised CBX1 chromodomain), N-terminal mCherry tag, Constitutive CAG enhancer/promoter | This study |
| pEGFP-Human MeCP2 | Wild-type Human MeCP2, N-terminal EGFP tag, Constitutive CMV enhancer/promoter | 36 |
| pEGFP-Mouse MeCP2 | Wild-type Mouse MeCP2, N-terminal EGFP tag, Constitutive CMV enhancer/promoter | 95 |
| pEGFP-MeCP2 AT-Hookmut | Mouse MeCP2 AT-Hook mutant (R188G, R190G), N-terminal EGFP tag, Constitutive CMV enhancer/promoter | This study |
| pEGFP-MeCP2 ΔMBD | Mouse MeCP2 MBD mutant (Δ99–125), N-terminal EGFP tag, Constitutive CMV enhancer/promoter | This study |
| pEGFP-MeCP2 ΔMBD + AT-Hookmut | Mouse MeCP2 MBD and AT-Hook double mutant (Δ99–125, R188G, R190G), N-terminal EGFP tag, Constitutive CMV enhancer/promoter | This study |
| pEGFP-MeCP2 Minimal MBD | Mouse Minimal functional MBD (78–162), N-terminal EGFP tag, Constitutive CMV enhancer/promoter | This study |

A 'Two phase association' model (nonlinear regression) was used to fit experimental data using the software GraphPad Prism v9.5.1. The plateau (fluorescence at the last time point of the FRAP experiment), corresponding to the mobile fraction, was interpolated from the fitted curve. Conversely, the immobile fraction corresponds to the pool of fluorescent proteins not recovered during the experiment (=100% − mobile fraction). The $T_{50}$ (time to recover 50% of pre-bleach level) was interpolated from the fitted curve. Data was plotted using GraphPad Prism v9.5.1.

### Immunofluorescence

Immunofluorescence was performed as previously described in ref. 88. Cells were washed with PBS and fixed with 4% PFA for 10 min at room temperature. After fixation, cells were washed with PBS and permeabilised for 10 min at room temperature in PBS supplemented with 0.3% (v/v) Triton X-100. Samples were blocked for 1 h 30 min at room temperature in blocking buffer: PBS supplemented with 0.1% (v/v) Triton X-100, 1% (w/v) BSA and 3% (v/v) goat serum (ordered from Sigma-Aldrich). Following blocking, samples were incubated overnight at 4 °C with primary antibodies (see Table 3) diluted at the appropriate concentration in blocking buffer. After four washes in PBS supplemented with 0.1% (v/v) Triton X-100, samples were incubated for 2 h at room temperature (in the dark) with fluorescently labelled secondary antibodies (Invitrogen Alexa Fluor Plus antibodies) diluted (1:500) in blocking buffer. Cells were washed four times with PBS supplemented with 0.1% (v/v) Triton X-100. DNA was stained with 1 μg/ml DAPI diluted in PBS for 5 min at room temperature, and cells were submitted to a final wash with PBS. Samples were mounted on coverslips using the ProLong Glass mounting medium (Thermo Fisher Scientific cat. P36980) and imaged using the Zeiss LSM 880 confocal microscope (with Airyscan module). Images were acquired using the Zeiss ZEN software (black edition) and processed using the software Fiji.

For 5-methylcytosine immunostaining, we performed the same protocol as described above with extra steps. Following permeabilization, DNA was denatured with 4 M HCl for 10 min at 37 °C. The pH of the

medium was subsequently neutralised with three quick washes in PBS supplemented with 0.1% (v/v) Triton X-100. We then proceeded with the blocking step and followed the rest of the immunostaining protocol.

For brain immunostaining, frozen tissue was sectioned into 10 μm sections using a Cryostat (Leica CM1900) and mounted onto Superfrost Plus microscope slides. Sections were air-dried briefly and then stored immediately at −80 °C. Brain sections were thawed and allowed to dry completely before being fixed and permeabilised by incubating with pre-cooled methanol:acetone (1:1) mixture for 20 min at −20 °C. Sections were washed in distilled water for 5 min, followed by PBS for 5 min. Slides were blocked in 1.5% goat serum for 20 min and then incubated for ≈1 h with MeCP2 antibody (see Table 3) diluted in 2.5% goat serum before being washed once in PBS for 10 min and incubated with biotinylated secondary antibody (Vector Laboratories cat. PK-6101) for 30 min at room temperature. Sections were washed again once in PBS for 10 min before being incubated for 15 min at room temperature with Fluorescein Avidin DCS (Vector Laboratories cat. A-2011) diluted 1:200 in 10 mM HEPES pH7.9, 150 mM NaCl solution. Slides were washed once in PBS for 10 min, incubated for 10 min in 1 μg/ml DAPI diluted in PBS, and washed again twice in PBS for 10 min. Samples were mounted in ProLong Glass mounting medium (Thermo Fisher Scientific cat. P36980) and imaged using the Zeiss LSM 880 confocal microscope (with Airyscan module). Images were acquired using the Zeiss ZEN software (black edition) and processed using the software Fiji.

### Fluorescence in situ hybridisation (FISH)

DNA FISH was performed as indicated in previous studies[55,89] with minor adaptations. First, immunofluorescence for MeCP2 was performed as indicated above, followed by a post-fixation in 4% PFA for 10 min at room temperature. Following three washes in PBS, cells were treated with 0.1 mg/ml RNase A (diluted in PBS) for 1 hr at 37 °C. After three washes in PBS, samples were incubated for 10 min on ice in PBS supplemented with 0.7% (v/v) Triton X-100 and 0.1 M HCl. Cells were washed three times in PBS to neutralise the pH, followed by a

denaturation in 2x SSC/50% formamide for 30 min at 80 °C. After three washes in ice-cold PBS, samples were incubated overnight at room temperature in hybridisation mix (2x SSC, 60% formamide, 5 µg/ml herring sperm DNA and 5 nM fluorescently labelled probe). The following probe was used for staining major satellite DNA: 5′-Cy5-GGAAAATTTAGAAATGTCCACTG-3′. Cells were washed two times for 15 min at room temperature in 2x SSC/50% formamide, and two times for 15 min at 42 °C in 2x SSC. Finally, samples were mounted on coverslips in Vectashield® medium containing DAPI (Vector Laboratories cat. H-1200) and stored at 4 °C until imaging. Samples were imaged using the Zeiss LSM 880 confocal microscope (with Airyscan module). Images were acquired using the Zeiss ZEN software (black edition) and processed using the software Fiji.

### Western blot

Western blot analysis was performed as previously described in ref. 41. To prepare nuclear protein extracts, 10 million cells were lysed for 30 min at 4 °C in hypotonic buffer (10 mM HEPES pH7.9, 5 mM $MgCl_2$, 0.25 M sucrose, 0.1% NP-40) supplemented with protease inhibitors (Roche). Nuclei were isolated by centrifugation and washed in lysis buffer. Acid extraction of histones[90] was performed in 0.4 M HCl overnight at 4 °C, followed by centrifugation to discard insoluble material and by neutralisation with NaOH. SDS-PAGE and protein transfer were performed according to standard procedures. Blots were probed with antibodies against total Histone H3 (Abcam cat. ab1791, 1:40,000 dilution), H3K9me1 (Active Motif cat. 39681, 1:2,500 dilution), H3K9me2 (Jenuwein Laboratory #4677, 1:5000 dilution) and H3K9me3 (Abcam cat. ab8898, 1:20,000 dilution).

### Southern blot

Genomic DNA was extracted using a commercial kit (Qiagen cat. 158388) and Southern blot was performed as previously described[91].

Equal amounts of genomic DNA were digested and loaded into each lane. Mouse (*M. musculus* and *M. spretus*) genomic DNA was digested with methylation-sensitive (HpyCH4IV, NEB cat. R0619L) or methylation-insensitive (ApoI, NEB cat. R3566L) restriction enzymes cutting within major satellite DNA repeats. Red deer genomic DNA was digested with methylation-sensitive (HpaII, NEB cat. R0171M) or methylation-insensitive (MspI, NEB cat. R0106M) restriction enzymes recognising the same 'CCGG' sequence within satellite I DNA. For detection of mouse major satellite DNA, we used a 804 bp probe (see Table 4) including three repeats of the consensus sequence[92], which was PCR-amplified from genomic DNA and cloned into a plasmid. For detection of red deer satellite DNA, we used a 725 bp probe (see Table 4) corresponding to the satellite I consensus sequence (GenBank accession numbers: U48429, MT185963, MT185964, MT185965, MT185966)[56,58], which was ordered as a synthetic double-stranded DNA fragment from IDT. Southern blot data was acquired using the Typhoon FLA 7000 Control software.

### 5-methylcytosine mass spectrometry quantification (LC-MS)

Genomic DNA from two independent biological replicates was extracted using a commercial kit (Qiagen cat. 69504) with an optional RNase A treatment step. The quantification of global 5-methylcytosine levels was performed by Zymo Research (US) using an SRM-based mass spectrometry assay. Samples were measured in technical triplicates and data were represented as total amount of 5-methylcytosine relative to the total pool of guanine within genomic DNA (%mC/G). Data plotted on GraphPad Prism v9.5.1.

### Protein alignment

MeCP2 protein sequences of the mammalian species used in this study were obtained from the UniProt database (UniProt identifiers: Q9Z2D6 (Mouse), P51608 (Human), Q00566 (Rat), A0A3Q2HTZ1 (Horse),

---

**Table 3 | List of primary antibodies used in the study**

| Antibodies | | |
|---|---|---|
| **Target** | **Working dilution** | **Reference** |
| H3K9me3 (Rabbit polyclonal antibody) | 1:200 | Abcam cat. ab8898 |
| HP1α (Rabbit polyclonal antibody) | 1:200 | Cell Signalling cat. 2616 |
| MeCP2 (Rabbit monoclonal antibody) | 1:200 | Cell Signalling cat. 3456 |
| TUJ1 (TUBB3) (Mouse monoclonal antibody) | 1:500 | BioLegend cat. 801201 |
| 5-methylcytosine (Mouse monoclonal antibody) | 1:1,000 | Active Motif cat. 39649 |

---

**Table 4 | Sequence of probes used for Southern blot**

| | |
|---|---|
| Red deer Satellite I probe (Southern blot) | CAAGACGAAAGGATGTCTGAATCCCCTGTGGAGACCACAGAGAAAGACCTAGTTCCCCACCTCATCGCGACCGGAGGCCTCA-CATCCTTTGAAAACTCCAGAGGTACGCGGAGATCAGTGCCTCCAAAGGAGACGATGCCTGACTCCTCGTGAAACTTGA-TAGGAGTCCCAGGATTCCTGTGGCACGTGGAAAGGGACCCTTGGTCTCCCGCCTCAGCTGGA-GAGGCGTCCCAATTGCCCTGCCAAGCCTCGAGGAGAATCCCGAGTTGTCCCTCGCAACTAGGCAGGAGTCCTGACGTCGCTGAAGAAA-CACGTGTGTGGAAGGGCCCATCCCCGTCGTAACTCGAGAATATACCCCAGGTTCCCGCCCGCAACTCGAGAAAAACCATGA-GACTTCCCCCTCGCCGCGAGATGAGGCCCGATTCCCCTGCACTGCGTGCAGAGCAATTCCGTGTTGCACATCACA-CATGAAAGGAGCCTTGATTTCCTTGATGGCACTCCAGAGAAACCCCAAGAACACTGTTTCAAGGCTA-GAGGGATCCTGAGGTCACTGTAGCAACACGAAAGAGCTCCGTGGACCAAAAATCAACTCGAGATGA-GAGGTTAGTCCCTGGCTTCGACTCCAGAGGAATACCACCTTACCACAAGCACCTCAAGAGGAGGCTTCTCTCAGCTCTAGGTATGTGA-GAGGGACCCTGAGTTTGCGGCCTCAAGTGGAATGGACACC |
| Mouse Major satellite DNA probe (Southern blot) | CAGTGAGCGCGCGTAATACGACTCACTATAGGGCGAATTGGAGCTCCCGCGGTGCGGCCGCTCTA-GAACTAGTGGATCCCCCGGGCTGCAGCCCAATGTGGAATTCGCCCTTGGCGAGGAAAACTGAAAAAGGTGGAAAATTTA-GAAATGTCCACTGTAGGACGTGGAATATGGCAAGAAAACTGAAAATCATGGAAAATGAGAAATATCCACTTGACGACTTGAAAAATGA-CAAAATCCCTGAAAAACGTGAAAAATGAGAAATGCACACTGTAGGACCTGGAATATGGCGAGAAAACTGAAAATCACGGAAAATGAGAAA-TACACACTTTAGGATGTGAAATATGGCGAGGAAAACTGAAAAAGGTGGGAAATTTAGAAACGTCCACTGTAGGACGTGGAATATGGCAAGA-GAACTGAAAATCATGGAAAATGAGAAACATCCACTTGACGACTTGAAAAATGACGAAATCACTAAAAAACGTGAAAAATGAGAAATGCA-CACTGAAGGACCTGGAATATGGCGAGAAAACTGAAAATCACGGAAATGAGAGATACACACTTTAGGACGTGAAA-TATGGCGAGGAAAACTGAAAAAGGTGGAAAATTTAGAAATGTCCTGTGTAGGACGTGGAATATGGCAA-GAAAACTGAAAATCATGGAAAATGAGAAACATCCACTTGACGACTAAGGGCGAATTCCACAGTGGATATCAAGCTTATCGA-TACCGTCGACCTCGAGGGGGGGCCCGGTACCCAGCTTTTGTTCCCTTTAGTGAGGGTTAATTGCGCGCTTGGCGTAAT |

A0A8C2MED1 (Chinese hamster), M3WF10 (Cat), A0A8I3SAQ1 (Dog), A0A3Q1M2I1 (Cow), A0A8D1PU92 (Pig)). For other species for which a complete MeCP2 protein sequence was unavailable, we obtained transcript sequences (Ensembl ENSOART00020034580 (Sheep), NCBI XM_007993126 (African green monkey), NCBI XM_043895361 (Red deer)) and translated them in silico. All protein sequences were aligned using the Clustal Omega programme[93] available as an online tool on the UniProt website (https://www.uniprot.org/align). Alignment data were visualised using the software ESPRIPT[94].

### Reporting summary

Further information on research design is available in the Nature Portfolio Reporting Summary linked to this article.

## Data availability

Source data for Table 1, Figs. 1, 2, 3, 4 and associated Supplementary Figs. 1, 2, 3, 4, 5, 6, 7, 8, 9, 10 are provided with this paper (https://doi.org/10.5281/zenodo.8430514). This includes raw and processed microscopy data, uncropped gels and blots images and datasheets containing individual values underlying each plot.

## Materials availability

Critical reagents used in this study (plasmids, cell lines) are available upon request to Adrian Bird (a.bird@ed.ac.uk) with unrestricted access.

## Code availability

The custom script used for image segmentation and DNA foci quantification was deposited on Zenodo (https://doi.org/10.5281/zenodo.7740611). The script used for FRAP analysis was reported in a previous publication[9] and is available on Zenodo (https://doi.org/10.5281/zenodo.2654601).

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

## Acknowledgements

We thank Jacky Guy for advice and Verdiana Steccanella and Jenna Hare for technical assistance. We thank Ian Valentine, Simon Girling and Helen Senn (RZSS), Christine Tait-Burkard (Roslin Institute), Robbie Hunt (Moorfoot Estate), Craig Wilson (Corrie Fee Estate), and staff at Wishaw Abattoir for provision of Red River hog, warthog, deer and cow tissues. We also thank Gura Bergkvist, Xavier Donadeu, Bruce Whitelaw and Josephine Pemberton (University of Edinburgh) for sharing mammalian cell lines and genomic DNA used in this study, and Tuncay Baubec (Utrecht University) for sharing the chromodomain reporter construct. Imaging was performed in Centre Optical Instrumentation Laboratory supported by a Core Grant (203149) to the Wellcome Centre for Cell Biology at the University of Edinburgh. This work was funded by the European Research Council Advanced Grant EC 694295 Gen-Epix and Wellcome Investigator Award #107930. A.B. is a member and K.P. is supported by the Simons Initiative for the Developing Brain grant (SFARI-529085). The collection of primary cell lines from mammalian species was funded by a BBSRC Institute Strategic Programme grant BBS/E/D/10002071 to T.B. Research in the laboratory of T.J. is supported by the Max Planck Society and by grants (CRC992 'MEDEP') from the German Research Foundation (DFG).

## Author contributions

Conceptualization: A.B., R.P.; Methodology: R.P., M.B., S.H., K.P., T.B., S.M.; Software: T.Mc., D.A.K.; Validation: R.P., M.B., S.H.; Formal analysis: R.P.; Investigation: R.P., M.B., S.H., K.P., T.Mo.; Resources: T.B., S.M., T.Mo., N.S., T.J., T.H., C.L.; Writing—Original Draft: A.B., R.P.; Writing—Review & Editing: A.B., R.P., M.B., K.P., T.B., T.J., T.Mc.; Supervision: A.B., R.P.; Funding acquisition: A.B., T.B., T.J.

## Competing interests

The authors declare no competing interests.
