## [Peer Review File · Nature Communications]

MeCP2 binds to methylated DNA independently of phase separation and heterochromatin organisationEditorial Note: Parts of this Peer Review File have been redacted as indicated to remove third-party material where no permission to publish could be obtained.

REVIEWER COMMENTS

Reviewer #1 (Remarks to the Author):

In this study Pantier, Bird and colleagues investigate the localization of methyl-DNA binding protein MeCP2 in heterochromatin, assessing the role of the methyl-DNA binding capacity of the protein (via its methyl binding domain) compared to the liquid-liquid phase separation capacity of the protein (through its recently described intrinsically disordered domains e.g. Li et al. 2020) in this process. In particular, the authors investigate if the localization of MeCP2 to chromocenters (dense, hoechst staining DNA foci found in mouse nuclei), the formation of these chromocenters, and the colocalization of heterochromatin (HP1) at those centers, is dependent on these two activities of the protein. Through live cell imaging of tagged proteins and immunostaining analysis in multiple wild-type and DNA methylation/heterochromatin-deficient cell lines the authors convincingly show that methyl-DNA binding is necessary and sufficient for enriched chromocenter localization (via live cell imaging) as well as for increased residence time and stable binding of the protein in these regions (via FRAP). They show that disruption of one chromocenter-localized signal, MeCP2 or heterochromatin (via H3K9 methyltransferase mutation), does not impact the other signal or the formation of DNA foci, suggesting that formation of these foci is independent of MeCP2 condensation and heterochromatin. Through analysis of cells from numerous mammalian species (including related mouse species) the authors further show that the formation of chromocenters and the localization of MeCP2 and heterochromatin to these foci is not a conserved feature of mammalian cells, drawing into question importance of these sites for MeCP2 function. Overall, the study is clearly presented, the assays performed are high quality and well controlled, and the experimental findings are robust and convincing. In particular, the incorporation of multiple cell lines in the analysis supports each of the conclusions that are made.

Regarding the impact of the study, I feel that it addresses a key topic that will be of substantial interest to the field. Recent studies have detected phase-separating characteristics of MeCP2 in vitro, associating the localization of MeCP2 foci at chromocenters with this phase separation and suggesting functional role for MeCP2 in heterochromatin (Li et al. 2020, Fan et al. 2020). However, these studies have extrapolated the functional importance of the phase separation by MeCP2 to gene regulation and disease pathology purely through colocalization with chromocenters without any direct link between these centers and heterochromatin or gene expression. The results presented here suggest that phase separation and heterochromatin localization of MeCP2 need to be more stringently evaluated for their relevance to the function of the protein and to the pathology of Rett syndrome. This study is therefore highly relevant and timely for the field as it assesses the importance of the methyl-binding activity and proposed phase-separation function of MeCP2.

In sum, I find the study to be of high quality and high importance for the field. Before it is suitable for publication, however, there are some experimental and presentation issues that I feel are important to address:

Major issues:

-In several Figures (e.g. Figure 1B,C, Figure 2) boxplots are used to show the ranges of values observed in cells. While this presentation demonstrates robust effects across all cells analyzed it is not clear what the variability is between different experiments, and the n for the number of independent experiments performed is not clearly listed. This is particularly relevant because different transfection efficiencies between experiments or constructs could impact ratios of nuclear signal in cells. The authors should include the number of experiments performed and show per experiment averages on plots (rather than just the distributions of values across all cells without per experiment information). This is important to include to show that the values detected and conclusions drawn are reproducible across independent experiments. In addition, statistical significance is shown in many figures, but the

tests used to measure significance are not described in the figure legends. This information should be added.

-While the use of transfected constructs allows the authors to perform a number of well-controlled experiments, certain conclusions (such as the diffuse staining across species) would be better supported with some analysis of endogenous protein. Inclusion of immunostaining for endogenous protein (which should work given the high conservation of MeCP2 across species) would strengthen the findings.

- While the data in the manuscript make it clear that the MBD plays a major role in localization of MeCP2 to DNA foci in mouse cells, the data presented show that the MBD alone is not sufficient to fully localize it to these sites. Although the explanation that the AT-hooks may play a role is plausible, it is not clear to me that the authors can strictly rule out that IDR-based associations contribute to missing colocalization signal observed with the minimal MBD construct alone. Thus, it would be appropriate to more prominently acknowledge this potential minor contribution of the IDRs to this binding in the results or discussion.

-Given that the highest expression of MeCP2 and the presence of mCA (of a major binding site for the protein) only occur in mature neurons, it is a notable limitation of the study that no neurons that endogenously express in vivo-relevant levels of MeCP2 or contain significant mCA are examined (LUHMES neurons are immature). If the authors could add analysis of MeCP2 localization in adult mouse neurons in vivo (perhaps via viral transduction in both *Mus musculus* and *Mus spretus*) it would substantially strengthen the study. I will concede that, strictly speaking, as a follow-on investigation from recent studies of phase-separation by MeCP2 (e.g. Li et al. 2020, Fan et al. 2020) such an analysis in mature neurons is not necessary (since the previous studies in question have done little to no experimentation in mature cells). However, the more studies we publish that marginalize these substantial differences between the model cellular systems used and the disease relevant, mature neuron context in vivo, the more we risk making conceptual conclusions that have limited relevance to endogenous biology and disease.

Minor:

-The authors note that mCA is not present in the cells studied, but do not prominently discuss the relevant fact that mCA is also largely excluded from heterochromatin. This point is relevant to note as it further suggests that major activities of MeCP2 are in euchromatin.

-It is not clear why the construct schematics for proteins studied in Figure 1 are not shown in the main figure, and location of the GFP tag is also not included in the schematics throughout the paper. It would be helpful if this information is added.

-In the legend of figure S8 it would be helpful to clarify the color code description by changing "conservative substitutions found in several mammalian species are in red." to "conservative substitutions found in several mammalian species are in red text with white background".

Reviewer #2 (Remarks to the Author):

MeCP2 (Methyl-CpG-binding protein 2) is best known for its causal association with the severe neurological disorder, Rett Syndrome (RTT). A series of recent studies have shown that RTT-causing MeCP2 mutations disrupt the abilities of MeCP2-mediated liquid-like condensate formation of chromatin in vitro, which may underlie the pathology of MeCP2 deficiency-driven RTT. Using mostly cell-based analysis in combine with methylation-sensitive restriction enzyme digestion, the manuscript by Pantier et al. showed their evidence to complement the current understanding of how MeCP2

interact with heterochromatin regions, in particular those at chromocenters, in cultured mammalian cells. Based on the results, the authors claimed that MeCP2 lacks intrinsic tendency of forming condensates, raising the concern of whether MeCP2 could actually drive/promote heterochromatin formation in vivo via liquid-liquid phase separation (LLPS). While the results presented in this manuscript are interesting, most of the findings appear to be the reiteration of results reported before. Moreover, based on the current form of the manuscript, the interpretation of many results is not precise and sometimes misleading, which compromises the bases of the main conclusion. The manuscript needs to address several concerns listed below before it could be reconsidered as in a form for publication.

- The authors' perspective of whether MeCP2 could promote heterochromatin formation via LLPS primarily derived from two lines of evidences, a) whether the intrinsic disordered domains (IDRs) are required for the targeting of MeCP2 to heterochromatin regions (which, in the manuscript, appear to be used interchangeably with chromocenter regions by the authors), and b) whether MeCP2 could drive heterochromatin foci assembly de novo in the absence of a visible chromocenter organization within the nucleus. While MeCP2 could localize to and play a positive role in chromocenter organization, neither MeCP2 targeting nor the visual existence of the "atypical" chromocenters (like those in M.m. cells) needs to depend on MeCP2-regulated LLPS of chromatin. Therefore, the results showed in the manuscript are not contrary to what people already reported about MeCP2 (particularly those in vitro data about MeCP2-mediated LLPS), but actually complement our understanding in regards to MeCP2 functioning. The manuscript was written in a tone of refuting in vitro findings and concluded that MeCP2 has no intrinsic tendency of forming condensates in inappropriate. I recommend the authors to rephrase related parts (particularly in the Abstract and Results part) across the manuscript accordingly.

- The definition of heterochromatin is originally based on densely stained and seemingly compacted chromatin regions, which is almost identical to the regions that are labeled by DAPI or Hoechst in cytological assays. Besides, heterochromatin regions are also associated with other chromatin modifiers and epigenetic marks such as H3K9me2/3, H3K27me3 and H4K20me. In Figure 2B, the authors showed that the overall removal of H3K9me3 led to decreased binding of H3K9me3 readers including HP1 and the reporter CHD-mCherry. However, chromocenter regions are still intact under 5KO which may indicate that residual H3K9me or other heterochromatic marks still help to organize chromocenter regions. Therefore, it's not sure about whether heterochromatin has been already "dissolved" under this condition. More importantly, the role of MeCP2-mediated chromatin organization may play an even major role under 5KO as other heterochromatin organization factors were downregulated. This is actually testable by overexpressing MeCP2 mutants under 5KO and examine the organization of chromocenter organization.

- As the authors mentioned, the formation of chromocenters is not typical for heterochromatin organization. Indeed, the majority of mammalian cells does not possess any chromocenter-like structures in nuclei. It's not clear whether MeCP2 could play a regulatory role in heterochromatin regions in these cells. Similarly, the absence of chromocenter is not indicative of MeCP2 deficiency. In cases where densely stained nuclear foci could be observed (like in cat and Chinese hamster cells showed in Figure 3B), the authors could test if MeCP2-mediated LLPS actually contributes to the regulation of these foci, which are smaller than chromocenters in size but still easily distinguishable by microscopy.

- The organization of pericentromeric heterochromatin into chromocenters is a rather complex process which is regulated by a number of factors in a cooperative manner. Studies have shown that MeCP2 deletion does not eliminate chromocenters; rather, it triggers the feedback control loop which alters the internal properties of the chromocenters including histone PTMs and associated proteins within the regions (more see PMID: 26406379). Therefore, it's unlikely to "re-create" chromocenters by MeCP2 overproduction, even if MeCP2 shows LLPS-driving properties, under conditions described in the manuscript.

- Previous studies have shown that MBD plays an important role in targeting MeCP2 to chromocenters (PMID: 30157418). However, according to the authors' findings in the manuscript (Figures 1B-D & S3B-C) and studies from others, other parts (for example, the intervening domain (ID)) of MeCP2 also play important roles in MeCP2 targeting. Most importantly, other domains clearly regulate MBD-mediated MeCP2 targeting, as studies have shown that the MBD-containing mutant MeCP2-R168X fails to localize to chromocenters (PMIDs: 32111972 & 32698189). The authors argue that MBD is both necessary and sufficient for MeCP2 targeting is not accurate. Detailed functional analysis of MeCP2 truncations in both MeCP2 targeting and heterochromatin organization will be helpful. I suggest the authors to not only focus on the MBD domain but also analyze the contributions of other parts of the protein in their assay, which will significantly improve the manuscript.

Reviewer #3 (Remarks to the Author):

Comments on Pantier et al., MeCP2 binds to methylated DNA independently of phase separation and heterochromatin organisation.

In this manuscript the authors challenge the view that heterochromatin itself dictates MeCP2 organization and function, as a bona fide DNA methylation binding protein. They first use deletion constructs to show that in mouse cells MeCP2 localizes to DAPI-dense foci through its MBD and that a minimal MBD is sufficient for DNA-dense localization. In mouse cells devoid of all three DNMT enzymes, MeCP2 still largely localized at DNA-dense foci although its dynamic association with the condensed foci was dramatically reduced. They then test MeCP2 localization in cells which are devoid of five out of six enzymes responsible for H3K9 methylation. In these H3K9me3-less cells, DAPI-dense foci were largely unaffected, and MeCP2 localization also remained intact, suggesting that neither DNA methylation nor H3K9 methylation is responsible for MeCP2 localization at DAPI-dense foci. In fact, when the authors tested MeCP2 localization in cells from different species, they observed that the majority of the cells displayed diffuse MeCP2 localization, and that mouse (and red deer) are the exception, not the rule. Finally, comparing *Mus musculus*, which contains abundant major satellite DNA and *Mus spretus*, which does not, they report that MeCP2 localization is dependent on major satellite repeat transcription.

While potentially interesting, the paper has some serious drawbacks:

Major

1. The authors indicate that they used Super Resolution (SR) microscopy, but the images do not appear to be even remotely super-resolved, and nowhere in the paper can one find details regarding the SR method used (an Airy scan is not considered SR microscopy, for that matter). In the text (page 6) the authors refer to Figures 3B and C as "super-resolution" but in the figure legend "live cell imaging" is mentioned. Common SR microscopy are types of one out of the following three types of SR microscopy approaches:

- Fluorescence blinking-related super-resolved image reconstruction, such as DNA-PAINT, PALM, STORM or SOFI, which are usually performed in a wide-field setup, not in an LSM confocal microscope.
- Structured illumination-related super-resolved imaging.
- Laser scanning microscopy using a doughnut-shaped beam, such as STED or MINIFLUX.

None of these types were mentioned, and again, the images do not provide an indication of super-resolution.

2. Another issue pertains to the domain analysis the authors performed for MeCP2. The authors reach quite conclusive and far-reaching conclusions (i.e., that the MBD is necessary and sufficient for

MECP2's localization), but Figure 1C shows that the minimal MBD is in-between the WT and the delta-MBD, and that the delta-MBD itself does not abolish the localization, but rather reduces it from ~ 4.5 to around 2 foci per nucleus. Therefore, the modest reduction in binding efficiency that was reported for the Minimal MBD, hints towards an alternative (or perhaps more complex) story, and the controls performed are insufficient relative to the conclusions. It is also the case for the few missense mutations, which did not show an effect. Without testing more of the residues of both IDRs as well as other regions (which by themselves contain important features that assist in forming phase-separation-like behavior), the authors' conclusions are a bit of a stretch and are not entirely supported by the data, or lack thereof.

3. The authors nicely show that in the absence of DNA methylation, MeCP2 is highly mobile. This shows that DNA methylation is required for MeCP2's strong interaction, contrary to the authors' conclusions. Again, both in the images and in Figure 1E it is clear that MeCP2 still localizes to DNA-dense foci. This indicates that DNA methylation is not required for MeCP2's localization, but rather for its strong interaction (suggesting some 'locking' mechanism), and not "loss of DNA binding specificity" as the authors propose.

4. The "size buffering" experiments should be validated using other methods, such as half-bleach FRAP experiments or fluorescence anisotropy imaging.

5. Colocalization experiments are often hard to interpret, especially in cases where partial overlap exists between the two fluorescence channels, and also sometimes even full colocalization exists. Colocalization (partial/full) only suggests that there is a chance that it is due to interactions. However, in resolution-limited microscopy, a single image pixel shows the ensemble average of multiple emitters within $\sim 200 \times 200 \times 800$ nm volume (depending on the objective lens and the ex./em. Pinhole diameters), whereas a direct interaction between proteins is in proximities similar to the typical sizes of the proteins, hence within a few nanometers. FRET imaging, or real SR microscopy could have helped strengthen the strong claims more confidently than 2-color colocalization.

Point-by-point response to the reviewers' comments

We thank all three reviewers for taking the time to read our manuscript and for providing detailed feedback. We have added new experimental data and made modifications to the manuscript, as detailed below. We believe that this substantial revision provides a significant improvement and further strengthens the conclusions of our study.

- New experimental data

- 1- We transfected mutant cells lacking DNA methylation (DNMT TKO) with all our MeCP2 mutant constructs (New Supplementary Figure panels 2b and 2c).
- 2- We performed extra characterisation of 5KO fibroblasts, confirming complete loss of canonical heterochromatin marks H3K9me1/2/3 (New Supplementary Figure panels 3a, 3b and 3c).
- 3- We transfected mutant cells lacking heterochromatin (5KO) with all our MeCP2 mutant constructs. (New Supplementary Figure panels 4a and 4b).
- 4- We added guinea pig cells (JH4 clone 1) to our panel of mammalian species, also showing diffuse MeCP2 signal (New Figure panels 3a and 3c + New Supplementary Figure panel 6).
- 5- We added an extra figure panel showing representative images for the rare sub-population of cow and monkey cells with spotty MeCP2 signal (New Supplementary Figure panel 7a).
- 6- We performed immunostaining for endogenous MeCP2 in several of our mammalian cell lines (confirming our results from live-imaging experiments with transfected MeCP2) (New Supplementary Figure panel 7c).
- 7- We performed immunostaining for MeCP2 in mouse *versus* rat brain (cortex and hippocampus) (New Figure panel 3e + New Supplementary Figure panel 9b).
- 8- We performed Satellite DNA FISH combined with MeCP2 immunostaining in *M. musculus versus M. spretus* cells (New Figure panel 4b).
- 9- We performed additional replicate experiments for some of the figure panels to increase the number of analysed cells (New Figure panels 1c, 2g + New Supplementary Figure panel 5d).

- Other changes to the manuscript

- 1- For clarity, and to avoid redundancy, we merged Supplementary Figure panel 1a/b/c with Figure 1a/b/c to show all mutant MeCP2 constructs transfected in mouse fibroblasts (New Figure panels 1a, 1b, 1c).
- 2- We moved Figure S5D as a main Figure panel (New Fig. 2g), as these scatterplots provide a robust evaluation of the correlation between MeCP2 levels and chromocenters. Conversely, Fig. 2g (dividing MeCP2-expressing cells into 3 categories) moved into Supplementary material (New Supplementary Figure panel 5d).
- 3- For all quantifications, we indicated in Figure legend the number of analysed cells and independent replicate experiments, as well as information regarding statistical analysis (which is further detailed in the Materials and methods section).
- 4- We deposited all datasets generated in this study on Zenodo (<https://doi.org/10.5281/zenodo.8430514>). This includes raw and processed data used to generate the Figures.

For microscopy data: raw images, uncropped pictures and additional fields of view for each Figure panel.

For quantifications: All datasets divided by replicate experiments and datasheets containing individual values underlying each plot.

- 5- We deposited the custom script used for image segmentation and quantification of MeCP2 signal at DNA foci on Zenodo (<https://doi.org/10.5281/zenodo.7740611>). The script used for FRAP analysis was reported in a previous publication, and is also available on Zenodo (<https://doi.org/10.5281/zenodo.2654601>).
- 6- When known, we provided information regarding the sex of cell lines and animals. Of note, MeCP2 nuclear distribution is not influenced by sex, so this parameter was not considered in the design of our study.
- 7- We made changes to the text following reviewers' comment (detailed below) and changes to comply with Nature Communications formatting guidelines.

Please see below a point-by-point response to reviewer's comments.

Reviewer #1:

In this study Pantier, Bird and colleagues investigate the localization of methyl-DNA binding protein MeCP2 in heterochromatin, assessing the role of the methyl-DNA binding capacity of the protein (via its methyl binding domain) compared to the liquid-liquid phase separation capacity of the protein (through its recently described intrinsically disordered domains e.g. Li et al. 2020) in this process. In particular, the authors investigate if the localization of MeCP2 to chromocenters (dense, hoechst staining DNA foci found in mouse nuclei), the formation of these chromocenters, and the colocalization of heterochromatin (HP1) at those centers, is dependent on these two activities of the protein. Through live cell imaging of tagged proteins and immunostaining analysis in multiple wild-type and DNA methylation/heterochromatin-deficient cell lines the authors convincingly show that methyl-DNA binding is necessary and sufficient for enriched chromocenter localization (via live cell imaging) as well as for increased residence time and stable binding of the protein in these regions (via FRAP). They show that disruption of one chromocenter-localized signal, MeCP2 or heterochromatin (via H3K9 methyltransferase mutation), does not impact the other signal or the formation of DNA foci, suggesting that formation of these foci is independent of MeCP2 condensation and heterochromatin. Through analysis of cells from numerous mammalian species (including related mouse species) the authors further show that the formation of chromocenters and the localization of MeCP2 and heterochromatin to these foci is not a conserved feature of mammalian cells, drawing into question importance of these sites for MeCP2 function. Overall, the study is clearly presented, the assays performed are high quality and well controlled, and the experimental findings are robust and convincing. In particular, the incorporation of multiple cell lines in the analysis supports each of the conclusions that are made.

Regarding the impact of the study, I feel that it addresses a key topic that will be of substantial interest to the field. Recent studies have detected phase-separating characteristics of MeCP2 in vitro, associating the localization of MeCP2 foci at chromocenters with this phase separation and suggesting functional role for MeCP2 in heterochromatin (Li et al. 2020, Fan et al. 2020). However, these studies have extrapolated the functional importance of the phase separation by MeCP2 to gene regulation and disease pathology purely through colocalization with chromocenters without any direct link between these centers and heterochromatin or gene expression. The results presented here suggest that phase separation and heterochromatin localization of MeCP2 need to be more stringently evaluated for their relevance to the function of the protein and to the pathology of Rett syndrome. This study is therefore highly relevant and timely for the field as it assesses the importance of the methyl-binding activity and proposed phase-separation function of MeCP2.

Indeed, our aim was to question the nature of the relationship between MeCP2, heterochromatin and chromocenters in live cells (simple correlation or functional interplay?), and to investigate the mechanistic basis of MeCP2 focal localisation in mouse cells (phase separation or biochemical affinity for methylated DNA?). We are happy that this reviewer finds our experimental data convincing, and we thank him/her for this very positive feedback.

In sum, I find the study to be of high quality and high importance for the field. Before it is suitable for publication, however, there are some experimental and presentation issues that I feel are important to address:

Major issues:

-In several Figures (e.g. Figure 1B,C, Figure 2) boxplots are used to show the ranges of values observed in cells. While this presentation demonstrates robust effects across all cells analyzed it

is not clear what the variability is between different experiments, and the *n* for the number of independent experiments performed is not clearly listed. This is particularly relevant because different transfection efficiencies between experiments or constructs could impact ratios of nuclear signal in cells. The authors should include the number of experiments performed and show per experiment averages on plots (rather than just the distributions of values across all cells without per experiment information). This is important to include to show that the values detected and conclusions drawn are reproducible across independent experiments. In addition, statistical significance is shown in many figures, but the tests used to measure significance are not described in the figure legends. This information should be added.

For all quantifications (box plots, scatterplots, line graphs), we have added in Figure legend the number of analysed cells, the number of independent replicate experiments ($n \geq 2$) and the statistical test used for analysis (additional details are provided in the Methods section): see Fig. 1c, 1e, 1f, 2c, 2d, 2g, 3c, 4f and Supplementary Fig. 1b, 2a, 2c, 4b, 4d, 5d. High resolution live-cell imaging is time-consuming and relatively low-throughput (usually 1-2 transfected cells per field of view). During our experiments, we aimed to image $\geq 10 \times$ fields of view for each condition to obtain a representative result of the global cell population.

We observed very consistent results between independent transfection experiments (see below examples for MeCP2 mutant transfections in 3T3 cells presented in Fig. 1c, and for FRAP analysis presented in Fig. 1f). We have pooled replicate experiments to increase the number of analysed cells, and therefore to increase the power of our statistical analyses. Furthermore, our MeCP2 levels analysis (Fig. 2e, 2f, 2g, Supplementary Figure panel 5d) showed that the ratio of MeCP2 fluorescence at DNA-dense foci vs nucleoplasm, which is the parameter measured in all our box plots, is relatively stable and not affected by varying MeCP2 expression levels.

Additionally, we have deposited all the datasets generated in this study on Zenodo (see link on page 1-2). This includes all results separated by independent replicate experiments, and datasheets with values underlying each plot.

-While the use of transfected constructs allows the authors to perform a number of well-controlled experiments, certain conclusions (such as the diffuse staining across species) would be better supported with some analysis of endogenous protein. Inclusion of immunostaining for endogenous protein (which should work given the high conservation of MeCP2 across species) would strengthen the findings.

While MeCP2 is expressed at low levels in fibroblasts (compared to neuronal cells), we could detect endogenous MeCP2 in our mouse (3T3), cow, roe deer and warhog cell lines (see New Supplementary Fig. 7c). These results obtained by immunostaining in fixed cells agree with our live imaging data with transfected EGFP-MeCP2 (Fig. 3b, 3c and Supplementary Fig. 6). Furthermore, our data in LUHMES cells shows live imaging of endogenous MeCP2 in human post-mitotic neurons (Fig. 3d), as these cells have been engineered by CRISPR/Cas9 to tag the endogenous locus with mCherry (see Shah et al, 2016 for more details, cited in this manuscript). Additionally, we have performed MeCP2 immunostaining from mouse and rat brain tissue (see New Fig. 3e and New Supplementary Fig. 9b). Collectively, this data on endogenous MeCP2, together with our live-imaging data with transfected MeCP2, shows that MeCP2 nuclear distribution is diffuse and chromocenters are not detectable in most mammalian species.

- While the data in the manuscript make it clear that the MBD plays a major role in localization of MeCP2 to DNA foci in mouse cells, the data presented show that the MBD alone is not sufficient to fully localize it to these sites. Although the explanation that the AT-hooks may play a role is plausible, it is not clear to me that the authors can strictly rule out that IDR-based associations contribute to missing colocalization signal observed with the minimal MBD construct alone. Thus, it would be appropriate to more prominently acknowledge this potential minor contribution of the IDRs to this binding in the results or discussion.

We acknowledge that the MBD is necessary, but not completely sufficient for robust localisation of MeCP2 to DNA-dense foci in mouse cells. We amended the text in our Results and Discussion sections to reflect this point:

- MeCP2 mutants analysis in wild-type cells (Results section): “Deletion of 27 amino acids within the MBD (Δ 99-125) greatly decreased co-localisation with DNA-dense foci, causing instead accumulation of MeCP2 in larger non-overlapping nuclear bodies (Fig. 1b). Therefore the two proposed intrinsically disordered regions (IDRs) of MeCP2 were unable to target MeCP2 to heterochromatin in the absence of a functional MBD. To test whether the MBD was sufficient for correct subnuclear localisation we expressed an EGFP-tagged 85 amino acid peptide corresponding to the minimal MBD. Despite lacking the two intrinsically disordered regions of MeCP2 (IDR1 and IDR2), the minimal MBD localised to DNA-dense foci (Fig. 1b). To quantify this distribution, we compared the intensity of fluorescence within foci versus the remaining nucleoplasm. This analysis confirmed the preference of the MBD for DNA-dense foci, suggesting that the MBD is sufficient as well as necessary for targeting heterochromatic foci. **This conclusion is qualified by the observation that the level of focal fluorescence is significantly lower than that of full length MeCP2 (Fig. 1c). This may be due to the proximity of the relatively large EGFP tag, but it is also possible that other MeCP2 regions play a role in stabilising binding to chromocenters.** For example, the minimal MBD lacks a nuclear localisation signal, which is not essential for MeCP2 entry into the nucleus, but may increase binding by raising its nuclear concentration. Additionally, DNA binding specificity may be provided by three potential AT Hooks, of which only AT Hook1 shows a marked preference for AT-rich DNA in vitro (Fig. 1a), although mutation of AT Hook1 (R188G, R190G) revealed no detectable contribution to MeCP2 sub-nuclear localisation either in the presence or absence of a functional MBD (Fig. 1a, 1b, 1c). These findings in fibroblasts were replicated in mouse embryonic stem cells (ESCs) (Supplementary Fig. 1a, 1b). Taken together, the evidence indicates that **the MBD is strictly necessary for heterochromatic localisation, but we cannot rule out that other regions of the protein, including perhaps “intrinsically disordered regions”, contribute to robust occupation of these sites.**”
- MeCP2 mutants analysis in DNMT TKO (Results section): “Inactivation of AT Hook 1 had no effect on this nuclear distribution. **Neither the MBD alone nor full-length MeCP2 lacking a functional MBD targeted chromocenters in DNMT TKO ESCs (Supplementary Fig. 2b, 2c), indicating that the full-length protein is required for heterochromatic localisation in the absence of DNA methylation.**”
- MeCP2 mutant analysis in 5KO (Results section): “**Similar to our results in wild-type cells, the MBD is critical for proper localisation of MeCP2 to chromocenters in 5KO, but the minimal MBD alone showed reduced binding** (Supplementary Fig. 4a, 4b).”
- Discussion section: “Our data are compatible with a simple biochemical explanation for nuclear localisation of MeCP2 based on its affinity for methylated DNA. **The 85 amino acid MBD of MeCP2 alone is sufficient to target DNA-dense foci in mouse cells, although with lower efficiency compared to the full-length protein.** Interestingly, the minimal MBD localises to chromocenters only in the presence of DNA methylation, while mutation of the MBD within the full-length protein abolishes MeCP2 subnuclear localisation.”

-Given that the highest expression of MeCP2 and the presence of mCA (of a major binding site for the protein) only occur in mature neurons, it is a notable limitation of the study that no neurons that endogenously express in vivo-relevant levels of MeCP2 or contain significant mCA are examined (LUHMES neurons are immature). If the authors could add analysis of MeCP2

localization in adult mouse neurons in vivo (perhaps via viral transduction in both *Mus musculus* and *Mus spretus*) it would substantially strengthen the study. I will concede that, strictly speaking, as a follow-on investigation from recent studies of phase-separation by MeCP2 (e.g. Li et al. 2020, Fan et al. 2020) such an analysis in mature neurons is not necessary (since the previous studies in question have done did little to no experimentation in mature cells). However, the more studies we publish that marginalize these substantial differences between the model cellular systems used and the disease relevant, mature neuron context in vivo, the more we risk making conceptual conclusions that have limited relevance to endogenous biology and disease.

We agree that specific accumulation of mCA in mature neurons is biologically important, and certainly impacts how MeCP2 regulates gene expression. However, the nuclear pattern of MeCP2 observed by microscopy seems to be conserved across cell types. In mouse, MeCP2 localises robustly to chromocenters in neurons, fibroblasts (as used in this study) and other specialised cell types in a variety of peripheral tissues including intestine, liver, kidney, muscle and skin (see Song et al, 2014). Although cultured cell lines do not accumulate mCA, these allowed us to perform high resolution live-cell imaging experiments, which would be challenging to do in animals.

To address this reviewer's request for analysis of mature neurons, we have now compared immunostaining of MeCP2 in rat and mouse brain sections (see New Fig. 3e and New Supplementary Fig. 9b). Our results show that MeCP2 nuclear distribution is diffuse in mature rat neurons, both in the cortex and hippocampus. These results agree with our observations made by live-cell imaging on rat primary fibroblasts.

Minor:

-The authors note that mCA is not present in the cells studied, but do not prominently discuss the relevant fact that mCA is also largely excluded from heterochromatin. This point is relevant to note as it further suggests that major activities of MeCP2 are in euchromatin.

We have updated the Discussion section as follows: "Although MeCP2 localises prominently to heterochromatic foci in mouse, the protein is also bound genome wide to euchromatin, where mCG sites (and mCA in neurons) are highly abundant. While the relationship between MeCP2 binding to euchromatic genes and transcriptional regulation has been extensively characterised, the functional significance of MeCP2 at pericentric heterochromatin remains unclear."

-It is not clear why the construct schematics for proteins studied in Figure 1 are not shown in the main figure, and location of the GFP tag is also not included in the schematics throughout the paper. It would be helpful if this information is added.

All constructs are now presented in Fig. 1a. (we also merged previous Figure S1 with Fig. 1 to improve clarity and avoid redundancy in the presentation of data). We did not include the EGFP tag in the diagrams due to space constraints, but included this information in the Figure legend. For all Figures containing live imaging data, MeCP2 panels are clearly labelled as "MeCP2-EGFP".

-In the legend of figure S8 it would be helpful to clarify the color code description by changing "conservative substitutions found in several mammalian species are in red." to "conservative

substitutions found in several mammalian species are in red text with white background”.

We have updated this information accordingly.

Reviewer #2:

MeCP2 (Methyl-CpG-binding protein 2) is best known for its causal association with the severe neurological disorder, Rett Syndrome (RTT). A series of recent studies have shown that RTT-causing MeCP2 mutations disrupt the abilities of MeCP2-mediated liquid-like condensate formation of chromatin in vitro, which may underlie the pathology of MeCP2 deficiency-driven RTT. Using mostly cell-based analysis in combine with methylation-sensitive restriction enzyme digestion, the manuscript by Pantier et al. showed their evidence to complement the current understanding of how MeCP2 interact with heterochromatin regions, in particular those at chromocenters, in cultured mammalian cells. Based on the results, the authors claimed that MeCP2 lacks intrinsic tendency of forming condensates, raising the concern of whether MeCP2 could actually drive/promote heterochromatin formation in vivo via liquid-liquid phase separation (LLPS). While the results presented in this manuscript are interesting, most of the findings appear to be the reiteration of results reported before. Moreover, based on the current form of the manuscript, the interpretation of many results is not precise and sometimes misleading, which compromises the bases of the main conclusion. The manuscript needs to address several concerns listed below before it could be reconsidered as in a form for publication.

We thank the reviewer for taking the time to evaluate our manuscript. Although we disagree regarding the interpretation that MeCP2 could undergo phase separation *in vivo*, we hope that our detailed response and new experimental data will address some of his/her concerns. We believe that our study provides a significant improvement to our understanding of MeCP2 nuclear distribution, and goes beyond evidence using purified proteins *in vitro*. The suggestion that most of our data reiterates what has been published before is unjustified. In most published work, MeCP2 localisation was investigated by immunostaining in fixed cells. However, the addition of fixatives such as formaldehyde can cause artefacts by changing the sub-cellular localisation of proteins (Teves et al, 2016; Irgen-Giuro et al, 2022), and in particular for the case of MeCP2 (Schmiedeberg et al, 2009). Here, we circumvented this problem by performing high resolution live-cell imaging. Furthermore, to test mechanistic hypotheses, we investigated MeCP2 nuclear distribution in a variety of cellular contexts (MeCP2 mutant protein expression, genetically engineered mouse lines, cell lines from different mammalian species) and performed quantitative analyses. These experiments are novel and were not reported previously in the literature.

- The authors' perspective of whether MeCP2 could promote heterochromatin formation via LLPS primarily derived from two lines of evidences, a) whether the intrinsic disordered domains (IDRs) are required for the targeting of MeCP2 to heterochromatin regions (which, in the manuscript, appear to be used interchangeably with chromocenter regions by the authors), and b) whether MeCP2 could drive heterochromatin foci assembly de novo in the absence of a visible chromocenter organization within the nucleus. While MeCP2 could localize to and play a positive role in chromocenter organization, neither MeCP2 targeting nor the visual existence of the "atypical" chromocenters (like those in M.m. cells) needs to depend on MeCP2-regulated LLPS of chromatin. Therefore, the results showed in the manuscript are not contrary to what people already reported about MeCP2 (particularly those in vitro data about MeCP2-mediated LLPS), but actually complement our understanding in regards to MeCP2 functioning. The manuscript was written in a tone of refuting in vitro findings and concluded that MeCP2 has no intrinsic tendency of forming condensates in inappropriate. I recommend the authors to rephrase related parts (particularly in the Abstract and Results part) across the manuscript accordingly.

In our study, we provided several lines of evidence which conflict with a model of phase separation of MeCP2 in live cells:

- The MBD of MeCP2 is critical for localisation to DNA-dense foci in mouse cells, while the IDRs (reported to be important for phase separation of MeCP2) are largely dispensable. As described in page 6, we amended the text (Results and Discussion) to acknowledge that the MBD alone is not fully functional and probably requires other MeCP2 regions to stabilise its binding to chromocenters. However, a previous study from our laboratory showed that a radically truncated version of MeCP2 containing the MBD and NCoR-interaction domain, and lacking most IDRs, is fully functional in mouse models and can reverse a Rett-like phenotype (see Tillotson et al, 2017).
- We found no evidence of “concentration buffering” of MeCP2 in live cells, which is a critical feature of liquid-liquid phase separation (see Fig. 2e, 2f, 2g, Supplementary Figure panel 5d). Instead, our data argue for a model of “size buffering” meaning that when methylated sites at chromocenters are saturated, MeCP2 accumulates in the nucleoplasm.
- The formation of MeCP2 “condensates” is peculiar to mouse cells, as this property is not conserved in mammalian species, including human. We verified these findings in human post-mitotic neurons (LUHMES) *in vitro* (Fig. 3d) and in rat brain *in vivo* (new Fig. 3e and New Supplementary Fig. 9b), as MeCP2 physiological function is restricted to the central nervous system.

We propose that MeCP2 nuclear distribution is guided by DNA methylation patterns in the genome of the host cell. This model would fit with our experimental data, and explain opposing observations in mouse vs other mammalian species. In particular, we propose that clustering of highly methylated satellite DNA repeats could mediate the formation of chromocenters and lead to the occurrence of MeCP2 foci (Fig. 4g). We believe that our experiments comparing *Mus musculus* and *Mus spretus* strain strongly support this hypothesis (see New Fig. 4b).

- The definition of heterochromatin is originally based on densely stained and seemingly compacted chromatin regions, which is almost identical to the regions that are labeled by DAPI or Hoechst in cytological assays. Besides, heterochromatin regions are also associated with other chromatin modifiers and epigenetic marks such as H3K9me2/3, H3K27me3 and H4K20me. In Figure 2B, the authors showed that the overall removal of H3K9me3 led to decreased binding of H3K9me3 readers including HP1 and the reporter CHD-mCherry. However, chromocenter regions are still intact under 5KO which may indicate that residual H3K9me or other heterochromatic marks still help to organize chromocenter regions. Therefore, it's not sure about whether heterochromatin has been already “dissolved” under this condition. More importantly, the role of MeCP2-mediated chromatin organization may play an even major role under 5KO as other heterochromatin organization factors were downregulated. This is actually testable by overexpressing MeCP2 mutants under 5KO and examine the organization of chromocenter organization.

We have performed additional characterisation of our 5KO cell line, which lacks all “canonical” heterochromatin marks H3K9me1, H3K9me2 and H3K9me3 (see New Supplementary Fig. 3c). The additional observation that endogenous HP1 protein becomes diffuse (Supplementary Fig. 3e) further confirms the “dissolution” of heterochromatin in these cells. For more information regarding this line, please see Montavon et al, 2021.

We found no evidence that MeCP2 could modulate heterochromatin organisation as MeCP2 knockout cells retained heterochromatic foci (Supplementary Fig. 5b, 5c), and MeCP2 over-expression only led to a modest increase in the size of chromocenters which did not correlate with expression levels (see Fig. 2e, 2f, 2g and Supplementary Fig. 5d). To answer the specific point made by this reviewer, we did not observe any noticeable change in chromocenters organisation between untransfected 5KO cells and 5KO cells over-expressing MeCP2 (see representative images below from independent fields of view).

- As the authors mentioned, the formation of chromocenters is not typical for heterochromatin organization. Indeed, the majority of mammalian cells does not possess any chromocenter-like structures in nuclei. It's not clear whether MeCP2 could play a regulatory role in heterochromatin regions in these cells. Similarly, the absence of chromocenter is not indicative of MeCP2 deficiency. In cases where densely stained nuclear foci could be observed (like in cat and Chinese hamster cells showed in Figure 3B), the authors could test if MeCP2-mediated LLPS actually contributes to the regulation of these foci, which are smaller than chromocenters in size but still easily distinguishable by microscopy.

We did not observe any obvious chromocenters or MeCP2 foci in cat and chinese hamster cells (see our quantification of all transfected cells in Fig. 3c). However, we found that red deer cells (Fig. 3b) and a rare sub-population of cow and monkey cells (5 to 10% of the cells) showed MeCP2 foci reminiscent of mouse cells. We have added representative images of these cells in a New Supplementary Fig. 7a. This observation does not change our main conclusion that MeCP2 nuclear distribution is diffuse in most mammalian species. As we did not find convincing evidence for phase separated MeCP2 condensates in mouse cells, we did not feel the need to carry out additional characterisation on these additional species.

- The organization of pericentromeric heterochromatin into chromocenters is a rather complex

process which is regulated by a number of factors in a cooperative manner. Studies have shown that MeCP2 deletion does not eliminate chromocenters; rather, it triggers the feedback control loop which alters the internal properties of the chromocenters including histone PTMs and associated proteins within the regions (more see PMID: 26406379). Therefore, it's unlikely to "re-create" chromocenters by MeCP2 overproduction, even if MeCP2 shows LLPS-driving properties, under conditions described in the manuscript.

As detailed in the Discussion section, the changes in pericentric heterochromatin observed in MeCP2-null or overexpressing cells (chromocenter size, DAPI intensity, histone marks, etc.) are rather subtle (see Linhoff et al, 2015 and Ito-Ishida et al, 2020). Although some compensation mechanisms might occur, published evidence and data presented in this manuscript show that MeCP2 is largely dispensable for chromocenter formation and maintenance. Furthermore, we show in this study that chromocenters don't appear to be evolutionarily conserved, meaning that these structures may not be relevant for MeCP2 physiological function.

- Previous studies have shown that MBD plays an important role in targeting MeCP2 to chromocenters (PMID: 30157418). However, according to the authors' findings in the manuscript (Figures 1B-D & S3B-C) and studies from others, other parts (for example, the intervening domain (ID)) of MeCP2 also play important roles in MeCP2 targeting. Most importantly, other domains clearly regulate MBD-mediated MeCP2 targeting, as studies have shown that the MBD-containing mutant MeCP2-R168X fails to localize to chromocenters (PMIDs: 32111972 & 32698189). The authors argue that MBD is both necessary and sufficient for MeCP2 targeting is not accurate. Detailed functional analysis of MeCP2 truncations in both MeCP2 targeting and heterochromatin organization will be helpful. I suggest the authors to not only focus on the MBD domain but also analyze the contributions of other parts of the protein in their assay, which will significantly improve the manuscript.

In this study, we used wild-type MeCP2 and 4x different mutant constructs (Δ MBD, AT Hook mutant, Minimal MBD and AT Hook mutant + Δ MBD, see Fig. 1a). This allowed us to evaluate the relative importance of two well-characterised DNA binding domains (MBD and AT Hook1), and by extension the relevance of intrinsically disordered regions which were unaffected in these constructs. A small deletion within the MBD in the context of the full-length protein (Δ MBD) is sufficient to dramatically affect MeCP2 nuclear distribution (see Fig. 1b and Supplementary Fig. 1a), thereby demonstrating the critical importance of this functional domain. The minimal MBD is able to localise to chromocenters in mouse cells, although with lower efficiency (Fig. 1b, 1c and Supplementary Fig. 1a, 1b). **As described in page 6, we amended the text (Results and Discussion) to acknowledge that the MBD alone when fused to GFP is not fully functional and may require other MeCP2 regions to stabilise its binding to chromocenters.** Although non-MBD regions could play a functional role, this would only be accessory as none of them is sufficient to localise to chromocenters on their own.

Regarding MeCP2 R168X (MeCP2 with a large C-terminal truncation, but retaining the MBD domain), published evidence from our lab and others shows that this construct retains localisation to chromocenters (see Kumar et al, 2008 and Schmiedeberg et al, 2009 with published Figure shown below). This mutant appears to be particularly sensitive to formaldehyde fixation which dramatically affects its subcellular localisation observed by live cell imaging (Schmiedeberg et al, 2009).

[redacted]

Reviewer #3:

Comments on Pantier et al., MeCP2 binds to methylated DNA independently of phase separation and heterochromatin organisation.

*In this manuscript the authors challenge the view that heterochromatin itself dictates MeCP2 organization and function, as a bona fide DNA methylation binding protein. They first use deletion constructs to show that in mouse cells MeCP2 localizes to DAPI-dense foci through its MBD and that a minimal MBD is sufficient for DNA-dense localization. In mouse cells devoid of all three DNMT enzymes, MeCP2 still largely localized at DNA-dense foci although its dynamic association with the condensed foci was dramatically reduced. They then test MeCP2 localization in cells which are devoid of five out of six enzymes responsible for H3K9 methylation. In these H3K9me3-less cells, DAPI-dense foci were largely unaffected, and MeCP2 localization also remained intact, suggesting that neither DNA methylation nor H3K9 methylation is responsible for MeCP2 localization at DAPI-dense foci. In fact, when the authors tested MeCP2 localization in cells from different species, they observed that the majority of the cells displayed diffuse MeCP2 localization, and that mouse (and red deer) are the exception, not the rule. Finally, comparing *Mus musculus*, which contains abundant major satellite DNA and *Mus spretus*, which does not, they report that MeCP2 localization is dependent on major satellite repeat transcription.*

While potentially interesting, the paper has some serious drawbacks:

Major

1. The authors indicate that they used Super Resolution (SR) microscopy, but the images do not appear to be even remotely super-resolved, and nowhere in the paper can one find details regarding the SR method used (an Airy scan is not considered SR microscopy, for that matter). In the text (page 6) the authors refer to Figures 3B and C as “super-resolution” but in the figure legend “live cell imaging” is mentioned. Common SR microscopy are types of one out of the following three types of SR microscopy approaches:

- Fluorescence blinking-related super-resolved image reconstruction, such as DNA-PAINT, PALM, STORM or SOFI, which are usually performed in a wide-field setup, not in an LSM confocal microscope.*
- Structured illumination-related super-resolved imaging.*
- Laser scanning microscopy using a doughnut-shaped beam, such as STED or MINIFLUX.*

None of these types were mentioned, and again, the images do not provide an indication of super-resolution.

The Airyscan LSM880 used in this study qualifies as a “super-resolution” microscope, as this system provides a resolution of $\approx 120\text{nm}$ (see Huff et al, 2017; Wu and Hammer, 2021), which is beyond the limit of conventional optical microscopes ($\approx 250\text{nm}$). However, as this reviewer pointed out, several different super-resolution technologies have been developed and some of them reach a significantly better resolution (down to 20-50nm). To avoid confusion, we amended the text and referred to “high resolution” microscopy instead of “super-resolution” microscopy.

Our aim was to image chromocenters in live cells, which are relatively large objects with a diameter of 1 to $2\mu\text{m}$ (see Supplementary Figure panel 5d), and easily distinguishable using conventional microscopes. The Airyscan LSM880 setup allowed us to perform fast and high resolution imaging of cells transfected GFP-tagged constructs. Other technologies like STED or PALM provide a better resolution, but are difficult to combine with live-cell imaging and usually require the use of special fluorophores.

2. Another issue pertains to the domain analysis the authors performed for MeCP2. The authors reach quite conclusive and far-reaching conclusions (i.e., that the MBD is necessary and sufficient for MECP2's localization), but Figure 1C shows that the minimal MBD is in-between the WT and the delta-MBD, and that the delta-MBD itself does not abolish the localization, but rather reduces it from ~4.5 to around 2 foci per nucleus. Therefore, the modest reduction in binding efficiency that was reported for the Minimal MBD, hints towards an alternative (or perhaps more complex) story, and the controls performed are insufficient relative to the conclusions. It is also the case for the few missense mutations, which did not show an effect. Without testing more of the residues of both IDRs as well as other regions (which by themselves contain important features that assist in forming phase-separation-like behavior), the authors' conclusions are a bit of a stretch and are not entirely supported by the data, or lack thereof.

In our box plots, we quantified the ratio of MeCP2 fluorescence within foci versus nucleoplasm (not the number of foci per nucleus). This analysis showed that, although the minimal MBD localises properly to DNA-dense foci (Fig. 1b and Supplementary Fig. 1a), this construct shows a quantitative defect with decreased enrichment compared to full-length MeCP2. Δ MBD mutants not only show a dramatically decreased signal at DNA-dense foci, but they also accumulate at ectopic locations (see merged "DNA+MeCP2" signal in Fig. 1b and Supplementary Fig. 1a), which are not quantified in these box plots as they don't coincide with DNA foci.

Additionally, as described in page 6, we amended the text (Results and Discussion) to acknowledge that the MBD alone is not fully functional and may require other MeCP2 regions to stabilise its binding to chromocenters.

3. The authors nicely show that in the absence of DNA methylation, MeCP2 is highly mobile. This shows that DNA methylation is required for MeCP2's strong interaction, contrary to the authors' conclusions. Again, both in the images and in Figure 1E it is clear that MeCP2 still localizes to DNA-dense foci. This indicates that DNA methylation is not required for MeCP2's localization, but rather for its strong interaction (suggesting some 'locking' mechanism), and not "loss of DNA binding specificity" as the authors propose.

We agree with this reviewer that our experiments nicely show that DNA methylation is required for stable binding of MeCP2. However, this is not contrary to our conclusions, as this extract from the original text demonstrates: "*In agreement with a previous report, MeCP2 retained localisation to chromocenters in live DNMT TKO ESCs, although nucleoplasmic signal was also significantly elevated compared to the parental cell line (Fig. 1d, 1e). [...] In line with previous studies, wild-type ESCs showed incomplete fluorescence recovery, even >6 minutes after bleaching, whereas MeCP2 recovery was complete and rapid in DNMT TKO ESCs (Fig. 1f, Supplementary Fig. 2d). Incomplete recovery and failure of fluorescence to plateau in wild-type cells prevents simple numerical comparison between wild-type and mutant cells, but it is evident that the time to reach 50% recovery is greatly reduced in DNMT TKO cells (~10 seconds versus ~75 seconds; Table 1). The data show that the stably bound fraction of MeCP2 is abolished in the absence of DNA methylation and MeCP2 binding becomes much more transient and dynamic, suggesting a loss of DNA binding specificity.*"

We have now transfected all our MeCP2 mutant constructs in DNA methylation-deficient cells (see New Supplementary Fig. 2b, 2c), highlighting again the importance of the MBD for chromocenter localisation. The absence of high affinity DNA binding sites for MeCP2 (mCG) in DNMT TKO ESCs is the most likely explanation for our experimental observations (lower

enrichment at chromocenters and transient chromatin binding), which is why we proposed that this represents a loss of DNA binding specificity.

4. The “size buffering” experiments should be validated using other methods, such as half-bleach FRAP experiments or fluorescence anisotropy imaging.

The half-bleach FRAP assay was already performed for MeCP2 and HP1 in immortalised mouse embryonic fibroblasts (Erdel et al, 2020), showing no preferential internal mixing within chromocenters. As we used a similar cellular system (3T3 cells), we did not consider it necessary to repeat this experiment. We cited this published study in the Discussion section of our manuscript: “*Our conclusions agree with a previous study that used live imaging approaches (including the half-bleach FRAP assay) to show that the proteins HP1 and MeCP2 do not behave as members of a phase-separated compartment in mouse cells.*”

5. Colocalization experiments are often hard to interpret, especially in cases where partial overlap exists between the two fluorescence channels, and also sometimes even full colocalization exists. Colocalization (partial/full) only suggests that there is a chance that it is due to interactions. However, in resolution-limited microscopy, a single image pixel shows the ensemble average of multiple emitters within ~200x200x800 nm volume (depending on the objective lens and the ex./em. Pinhole diameters), whereas a direct interaction between proteins is in proximities similar to the typical sizes of the proteins, hence within a few nanometers. FRET imaging, or real SR microscopy could have helped strengthen the strong claims more confidently than 2-color colocalization.

Here, we investigated MeCP2/heterochromatin concentration at chromocenters, which are large objects of 1-2 μ m diameter. We did not interpret co-localisation as a proxy for interactions between molecules, and this was not the aim of our study.

REVIEWER COMMENTS

Reviewer #1 (Remarks to the Author):

The authors have thoroughly addressed my concerns. The study will provide important new insights for the field that will garner significant interest. I believe it is now suitable for publication.

About the comments from Reviewer #3 and your responses

The issues raised by reviewer 3 asking whether the analysis in the study is true "super resolution" (point 1) and questioning whether they can assess co-binding by microscopy (point 5) do not seem particularly key or relevant to the assertions of the paper, and I feel that the authors make a convincing case that their text changes alone are sufficient. Points 2-4 raise more relevant issues, but the authors' responses are convincing. From my perspective, the changes made to the manuscript sufficiently address reviewer 3's critiques.

Reviewer #2 (Remarks to the Author):

The updated manuscript by Pantier et al. shows a collection of convincing results in addressing the concerns raised by me and others. The interpretation of their results has also been revised to be accurate. I have no more questions.

In large, I agree with the authors about the functional relevance between MeCP2-methylDNA binding and RTT pathology. Meanwhile, interesting observations in regards to certain RTT-causing MeCP2 mutations (for instance P389X, as documented in Li et al., 2020. Nature), which do have the integral MBD and NID while show defective LLPS, also call for additional work in the field for the full understanding mechanistically.

About the comments from Reviewer #3 and your responses

1. It's fair that the authors mainly rely on using Airyscan-acquired live-imaging data in the study, although technically a confocal microscopy is not generally categorized into any form of "super-resolution" microscopy. The authors revised their text from "super" to "high" should be enough to solve the confusion.
2. I agree with the reviewer that the conclusion of "MBD domain being both necessary and sufficient" is too bold, which is not well supported by the data. Particularly, the way the authors quantified the localization of MeCP2 truncations to chromocenters did not actually reflect the "patterning of MeCP2 in the nucleus" but rather the "stability of MeCP2 at chromosome centers". As partly mentioned in my comments to the manuscript, one can not rule out the contribution of domains other than MBD (especially the IDRs) to MeCP2 patterning. To this aim, the authors really need to either figure out a better way to quantify the pattern change of MeCP2 localization (for example, the numbers and diameters of the foci? Or any overlap coefficient between MeCP2 foci and chromocenter overlap, or similar sort), but not the amount of MeCP2 being at the chromocenters vesus within the nucleus. This will require the authors to re-quantify their image data. Another way to solve the concern is to further delete the IDRs out from delta MBD and compare MeCP2 patterning between delta (MBD) and delta (MBD+IDR). This will tell whether IDR contributes to MeCP2 patterning when MBD is absent. Because the study is providing an alternative view of MeCP2 phase separation, I think it's important to tease out this problem, which requires more than just text revision. Again this is up to the editor and authors to decide.
3. Similar to Point #2. One thing I want to add to this comment is that, in DNMT KO cells, MeCP2 remains foci-like, which could be due to the IDRs, but also could be due to the fact that MBD or other domains of MeCP2 prefer to bind to chromocenter DNA sequences without chromocenter DNA (major satellites) being methylated. Could be hydroxymethylated or other modifications, or not being modified at all. There are other studies which talked about the binding of MeCP2 MBD to many kinds of DNA sequences, with or without 5mC modifications. Of course, this will probably ask the authors to

add additional data which may not be appropriate to fit in a single study.

4. It's ok for the authors to cite papers that include the results the reviewer asked.

5. I see the points from both sides. I think that it's Ok for the authors to match "MeCP2 colocalizes to chromocenters" to "MeCP2 gets recruited to chromocenters", but if this approach becomes the only way to assess whether 5mC binding is sufficient to explain all MeCP2 behaviors, one needs to be careful, like what I believe in point #2 and #3.

Point-by-point response to the reviewers' comments

Reviewer #1

The authors have thoroughly addressed my concerns. **The study will provide important new insights for the field that will garner significant interest. I believe it is now suitable for publication.**

About the comments from Reviewer #3 and your responses

The issues raised by reviewer 3 asking whether the analysis in the study is true "super resolution" (point 1) and questioning whether they can assess co-binding by microscopy (point 5) do not seem particularly key or relevant to the assertions of the paper, and I feel that the authors make a convincing case that their text changes alone are sufficient. Points 2-4 raise more relevant issues, but **the authors' responses are convincing. From my perspective, the changes made to the manuscript sufficiently address reviewer 3's critiques.**

We thank this reviewer for his/her supportive feedback. We also believe that our work will have a significant impact in the field, and hopefully incite other scientists to have a more cautious approach when studying the potential phase separation of heterochromatin-binding proteins.

Reviewer #2

The updated manuscript by Pantier et al. shows a collection of **convincing results in addressing the concerns raised by me and others. The interpretation of their results has also been revised to be accurate. I have no more questions.**

In large, I agree with the authors about the functional relevance between MeCP2-methylDNA binding and RTT pathology. Meanwhile, interesting observations in regards to **certain RTT-causing MeCP2 mutations (for instance P389X, as documented in Li et al., 2020. Nature), which do have the integral MBD and NID while show defective LLPS, also call for additional work in the field for the full understanding mechanistically.**

We are happy that our point-by-point response and changes to the manuscript were considered as satisfying.

This reviewer's additional comments are focussed on chromocenters, which our paper shows are absent in almost all mammals. It follows that MeCP2 does not require chromocenter formation for its physiological function. As detailed in our study, we do recognise that non-MBD regions can significantly contribute to MeCP2 function (although the MBD is main driver for MeCP2 recruitment to chromocenters in mouse cells). The primary focus of our manuscript is to address the hypothesis that MeCP2 can undergo liquid-liquid phase separation (LLPS) *in vivo*. Our findings cast doubt on prior interpretation of the effects of Rett-causing mutations using reconstituted "droplets" *in vitro* (e.g. in Li et al, 2020). Indeed, our previous work showed that the MeCP2 mutation mentioned by this reviewer (P389X, which we termed "CTD2") retains localisation to chromocenters (Guy et al, 2018; see figure below). Rather than defective LLPS (as suggested by Li et al) we showed previously that decreased MeCP2 expression (to around 10-15% wild-type levels; see published Western-blot image below) is responsible for the pathogenicity of P389X mutation.

[redacted]

Adapted from Guy et al, 2018. (Left) Confocal imaging of EGFP-tagged MeCP2 transfected in mouse fibroblasts. (Right) Western blot analysis and quantification of endogenous MeCP2 expressed in wild-type and mutant human neurons.

About the comments from Reviewer #3 and your responses

1. It's fair that the authors mainly rely on using Airyscan-acquired live-imaging data in the study, although technically a confocal microscopy is not generally categorized into any form of "super-resolution" microscopy. The authors revised their text from "super" to "high" should be enough to solve the confusion.

2. I agree with the reviewer that the conclusion of "MBD domain being both necessary and sufficient" is too bold, which is not well supported by the data. **Particularly, the way the authors quantified the localization of MeCP2 truncations to chromocenters did not actually reflect the "patterning of MeCP2 in the nucleus" but rather the "stability of MeCP2 at chromosome centers"**. As partly mentioned in my comments to the manuscript, one can not rule out the contribution of domains other than MBD (especially the IDRs) to MeCP2 patterning. To this aim, the authors really need to either figure out a better way to quantify the pattern change of MeCP2 localization (for example, the numbers and diameters of the foci? Or any overlap coefficient between MeCP2 foci and chromocenter overlap, or similar sort), but not the amount of MeCP2 being at the chromocenters versus within the nucleus. This will require the authors to re-quantify their image data. Another way to solve the concern is to **further delete the IDRs out from delta MBD and compare MeCP2 patterning between delta (MBD) and delta (MBD+IDR)**. This will tell whether IDR contributes to MeCP2 patterning when MBD is absent. Because the study is providing an alternative view of MeCP2 phase separation, I think it's important to tease out this problem, which requires more than just text revision. Again this is up to the editor and authors to decide.

Our revised manuscript already discussed the potential contribution of non-MBD regions to chromocenter targeting in mouse cells (see previous rebuttal letter). If the IDRs contributed to the specificity for chromocenters we would expect the delta-MBD construct, which has a small inactivating deletion within the MBD and retains all IDRs, to localise to chromocenters. In fact this construct accumulates in blobs that are excluded from chromocenters (see Figure 1b, third row down).

The confusion appears to arise from our quantification method which involved image processing as described in the Materials & Methods section (our script is shared in a public database: <https://doi.org/10.5281/zenodo.7740611>). Our intention was to quantify mutant localisation relative to that of physiologically wild-type MeCP2, which efficiently co-localises

with chromocenters (DNA-dense foci) in mouse cells. To further clarify interpretation, we have added an extra Figure panel detailing image segmentation and quantification (New Supplementary Figure panels 1a). A score >1 indicates a relative enrichment of MeCP2 within chromocenters, a score =1 means no preference, and a score <1 means exclusion of MeCP2 from chromocenters. This unbiased approach allowed us to perform statistical analyses to compare the level of enrichment of MeCP2 between wild-type and mutant constructs, or between different cell lines. Therefore, we believe that our quantification method is correct, and reflects the conclusions drawn from our experiments. To the referee's suggestion that size and number of chromocenters could be measured, we respond that this was already reported in the manuscript and neither parameter correlates robustly with MeCP2 expression (see Figure 2g and Supplementary Figure 5d).

The suggestion that we generate a "delta (MBD+IDR)" construct is not feasible as almost all of the protein outside the MBD is classed as disordered. There would be virtually nothing left. As stated above, expression of Δ MBD (which comprises the entire IDR) shows convincingly that intrinsically disordered regions of MeCP2 can only play a minor role in directing MeCP2 to chromocenters in mouse cells (see Figure 1b, 1c).

3. Similar to Point #2. One thing I want to add to this comment is that, in DNMT KO cells, **MeCP2 remains foci-like, which could be due to the IDRs, but also could be due to the fact that MBD or other domains of MeCP2 prefer to bind to chromocenter DNA sequences without chromocenter DNA (major satellites) being methylated.** Could be hydroxymethylated or other modifications, or not being modified at all. There are other studies which talked about the binding of MeCP2 MBD to many kinds of DNA sequences, with or without 5mC modifications. Of course, this will probably ask the authors to add additional data which may not be appropriate to fit in a single study.

In agreement with this referee, our experiments showed that MeCP2 is preferentially at chromocenters in mouse cells in the complete absence of DNA methylation (*DNMT TKO* ESCs). However, despite this biased average subcellular localisation, FRAP analysis makes it clear that MeCP2 binds much more transiently to non-methylated chromocenters. Furthermore our mutation analysis showed that this DNA methylation-independent localisation to chromocenters requires the MBD but not the IDRs (see Supplementary Figure 2b, 2c). These results are compatible with low-affinity binding to non-methylated DNA sequences by the MBD (for example AT runs, see Klose et al, 2005) although its relevance to the physiological function of MeCP2 is questionable. We already acknowledge this possibility in the Results section as follows: "Taken together, the evidence indicates that the MBD is strictly necessary for heterochromatic localisation, but we cannot rule out that other regions of the protein, including perhaps "intrinsically disordered regions", contribute to robust occupation of these sites." In addition, we have now updated the Discussion further emphasising this point (see italicised text below). Note that hydroxymethylated cytosine (5hmC) cannot be responsible for residual binding, as suggested by this referee, as 5hmC is itself derived from 5mC (via oxidation by TET proteins) and is therefore completely absent in *DNMT TKO* ESCs.

"[...] As previously noted²⁴, MeCP2 retains its localisation to DNA-dense foci in mouse cells in the complete absence of DNA methylation (DNMT TKO ESCs), and this phenotype relies on an intact MBD domain. This could be explained by methylation-independent functions of the MBD, such as a weak intrinsic affinity for AT-rich sequences³⁶, or protein-protein interactions with heterochromatin-associated proteins such as ATRX⁷¹. However, the exchange rate of MeCP2 in DNMT TKO ESCs as measured by FRAP is greatly increased compared to wild-type cells, indicating reduced DNA binding affinity. It is notable that

*mutations that similarly reduce the residence time of MeCP2 on mouse heterochromatin cause Rett syndrome*⁶⁹, suggesting that fast exchange is incompatible with MeCP2 function.”

4. It's ok for the authors to cite papers that include the results the reviewer asked.

5. I see the points from both sides. I think that it's Ok for the authors to match "MeCP2 colocalizes to chromocenters" to "MeCP2 gets recruited to chromocenters", but if this approach becomes the only way to assess whether 5mC binding is sufficient to explain all MeCP2 behaviors, one needs to be careful, like what I believe in point #2 and #3.

We stress that our study aimed to investigate whether MeCP2 has the intrinsic propensity to form “condensates” *in vivo*, hence our live-cell imaging approach. Elucidation of other aspects of MeCP2 function (e.g. sequence specificity, protein-protein interactions, regulation of gene expression) has been the subject of numerous prior publications, as cited in the Introduction and Discussion sections, and was not a primary goal here.

REVIEWERS' COMMENTS

Reviewer #2 (Remarks to the Author):

The interpretation of their results has been revised to be accurate. I have no more questions.